# Genetic screens reveal a central role for heme metabolism in artemisinin susceptibility

Clare R. Harding[1,2,8], Saima M. Sidik[1,8], Boryana Petrova[1,8], Nina F. Gnädig[3], John Okombo[3], Alice L. Herneisen [1], Kurt E. Ward [3,4], Benedikt M. Markus [1,5], Elizabeth A. Boydston [1], David A. Fidock [3,6] & Sebastian Lourido [1,7✉]

Artemisinins have revolutionized the treatment of *Plasmodium falciparum* malaria; however, resistance threatens to undermine global control efforts. To broadly explore artemisinin susceptibility in apicomplexan parasites, we employ genome-scale CRISPR screens recently developed for *Toxoplasma gondii* to discover sensitizing and desensitizing mutations. Using a sublethal concentration of dihydroartemisinin (DHA), we uncover the putative transporter Tmem14c whose disruption increases DHA susceptibility. Screens performed under high doses of DHA provide evidence that mitochondrial metabolism can modulate resistance. We show that disrupting a top candidate from the screens, the mitochondrial protease DegP2, lowers porphyrin levels and decreases DHA susceptibility, without significantly altering parasite fitness in culture. Deleting the homologous gene in *P. falciparum*, *PfDegP*, similarly lowers heme levels and DHA susceptibility. These results expose the vulnerability of heme metabolism to genetic perturbations that can lead to increased survival in the presence of DHA.

[1] Whitehead Institute for Biomedical Research, Cambridge, MA, USA. [2] Wellcome Centre for Molecular Parasitology, Institute of Infection, Immunity & Inflammation, University of Glasgow, Glasgow, UK. [3] Department of Microbiology and Immunology, Columbia University Irving Medical Center, New York, NY, USA. [4] Department of Microbiology and Immunology, University of Otago, Dunedin, New Zealand. [5] Faculty of Biology, University of Freiburg, Freiburg, Germany. [6] Division of Infectious Diseases, Department of Medicine, Columbia University Irving Medical Center, New York, NY, USA. [7] Biology Department, Massachusetts Institute of Technology, Cambridge, MA, USA. [8] These authors contributed equally: Clare R. Harding, Saima M. Sidik, Boryana Petrova. ✉email: lourido@wi.mit.edu

Since their discovery and characterization as potent anti-malarials, artemisinin (ART), and its synthetic derivatives have emerged as important drugs in treating infectious diseases, and are being actively investigated in cancer therapy[1,2]. Notably, artemisinin-based combination therapies (ACTs) have transformed malaria treatment, particularly in response to widespread resistance to previous classes of antiparasitic compounds. ARTs require activation within cells by scission of an endoperoxide bridge through the $Fe^{2+}$ center of heme[1,3]. Cleavage of this bond produces ART radicals that react with a multitude of proteins[4,5], lipids[6], and metabolites[7], rapidly leading to cell death.

ARTs are labile, and their short half-lives (~1 h in humans) enable modest, stage-specific reductions in parasite susceptibility to translate into reduced clinical efficacy. This reduced activity was first noted in Southeast Asia a decade ago, and is increasingly frequent[8–11]. Clinical resistance has been associated with genetic polymorphisms in *Plasmodium falciparum* Kelch13 (K13), most notably K13$^{C580Y}$ [12,13]. Two recent publications attributed this decreased susceptibility to a hemoglobin import defect that presumably renders parasites less able to activate ART[14,15]. This hypothesis may help explain how mutants in other components of the endocytic machinery, such as coronin[16] and clathrin adaptors[17–19], and in hemoglobin digestion[20] also confer decreased susceptibility to ART. Previous studies have also postulated that the altered ART susceptibility of K13$^{C580Y}$ parasites involves changes in the unfolded protein response, autophagy, and PI3K activity, which may contribute to cellular survival following protein alkylation[21–24]. Some *P. falciparum* strains without identified causative mutations have also displayed decreased clearance[25,26], suggesting that aspects of ART susceptibility have yet to be elucidated.

Directed evolution coupled with whole-genome sequencing has identified targets and resistance pathways for many antiparasitic compounds[16,27–29]. Such approaches can be time-consuming, may fail to detect minor mutations or those that negatively impact parasite fitness, and can only be used for positive selection schemes. Recently, whole-genome CRISPR-based mutational approaches have uncovered drug-resistance mechanisms in cancer[30–32]. Analogously, we have shown that CRISPR screens in *Toxoplasma gondii* provide a comparatively fast method to identify causal mutations using both positive and negative selection[33,34].

Despite species-specific differences that could impact ART susceptibility—such as the lack of substantial hemoglobin uptake by *T. gondii*—here, we demonstrate that a point mutation in K13, homologous to the canonical *P. falciparum* K13$^{C580Y}$, reduced the susceptibility of *T. gondii* to dihydroartemisinin (DHA). Genome-wide screens in *T. gondii* further identified mutants in a putative porphyrin transporter (Tmem14c) that are more susceptible to DHA, as well as mutations in several genes involved in mitochondrial metabolism that decreased drug susceptibility. In particular, we identified a mitochondrial protease (DegP2), disruption of which decreases porphyrins and DHA susceptibility without impacting parasite fitness. We provide evidence that DegP2 interacts with several mitochondrial proteins, including one that contains a NifU domain that is predicted to be involved in transferring nascent iron–sulfur clusters to client proteins. In keeping with this observation, ΔDegP2 parasites have alterations in the electron transport chain (ETC) and the tricarboxylic acid cycle (TCA)—two iron–sulfur cluster-dependent processes.

## Results
### Using *T. gondii* to understand apicomplexan DHA susceptibility.
Previous studies have demonstrated the susceptibility of

*T. gondii* to ART and its derivatives[35–37]. To establish the susceptibility of *T. gondii* to DHA in our assays, we treated intracellular parasites with varying concentrations of DHA or pyrimethamine (the frontline treatment for toxoplasmosis) for 24 h. Parasite viability was measured by counting the number of vacuoles containing two or more parasites in vehicle- or drug-treated conditions. Both drugs completely blocked parasite replication at 10 μM, which did not affect the integrity of the host monolayers (Fig. 1a). Our results recapitulate the reported EC$_{50}$ for pyrimethamine[38] and underscore the potency of DHA against *T. gondii*. Nevertheless, we recognize that *T. gondii* is far less susceptible to DHA than blood-stage malaria parasites, a fact that contributes to the use of other compounds as frontline drugs for toxoplasmosis[39].

We next determined the length of time required for DHA to kill extracellular and intracellular parasites. Measuring the susceptibility of extracellular parasites to DHA allowed us to exclude possible effects of the compound on host cells. Isolated parasites were treated with DHA for increasing lengths of time before removing the compound and allowing parasites to infect host cells and replicate for 24 h. The effect of DHA (both at 1 μM and at 10 μM) plateaued after 5 h of treatment (Fig. 1b). We similarly determined the treatment window for intracellular parasites. We added 1 or 10 μM DHA to intracellular parasites 1 h post invasion (1 h.p.i.), washed out the compound after different lengths of time, then assessed parasite viability based on the proportion of the host cell monolayer that was lysed 72 h.p.i. (Fig. 1c). Based on these results, we used 5 h treatments for both intracellular and extracellular parasites[40].

### Homologous Kelch13 mutations decrease DHA susceptibility in *P. falciparum* and *T. gondii*.
In *P. falciparum*, point mutations in *Kelch13* (K13), such as C580Y and R539T, correlate with delayed clearance and increased survival of ring-stage parasites[12,13,41,42]. Although *K13* is conserved among apicomplexans, its role in DHA susceptibility has not been examined in *T. gondii*. We chose to make a C627Y mutation in the *T. gondii* ortholog of *K13* (TGGT1_262150), corresponding to *P. falciparum* C580Y[13] (Fig. 1d). Extracellular K13$^{C627Y}$ parasites were sevenfold less susceptible to DHA compared to the parental strain (Fig. 1e; EC$_{50}$ values provided in Supplementary Table 1), indicating that the effect of K13 on DHA susceptibility transcends genera. As in *P. falciparum*, K13$^{C627Y}$ parasites were still susceptible to prolonged DHA exposure and did not form plaques under continuous DHA treatment (Supplementary Fig. 1).

DHA susceptibility is known to change over the *P. falciparum* erythrocytic cycle, with trophozoites displaying greater susceptibility than rings or schizonts[43,44]. To determine whether active replication affects DHA susceptibility in *T. gondii*, we treated intracellular parasites immediately after invasion (Fig. 1f and Supplementary Table 1) before most parasites have initiated replication—or 24 h after invasion when most parasites are replicating[45] (Fig. 1g and Supplementary Table 1). K13$^{C627Y}$ parasites were at least twofold more resistant to DHA compared to the parental line, regardless of the timing of treatment (Fig. 1h), and we conclude that replication does not affect the reduced DHA susceptibility of K13$^{C627Y}$ *T. gondii*.

### A genome-wide screen identifies a mutant that is hypersensitive to DHA.
We performed CRISPR-based screens to identify genes that when disrupted would increase *T. gondii* susceptibility to DHA. Transfecting a library containing ten guide RNAs (gRNAs) per gene into a large population of parasites that constitutively expressed the Cas9 nuclease, we created a diverse population of mutants. From previous work, we know that

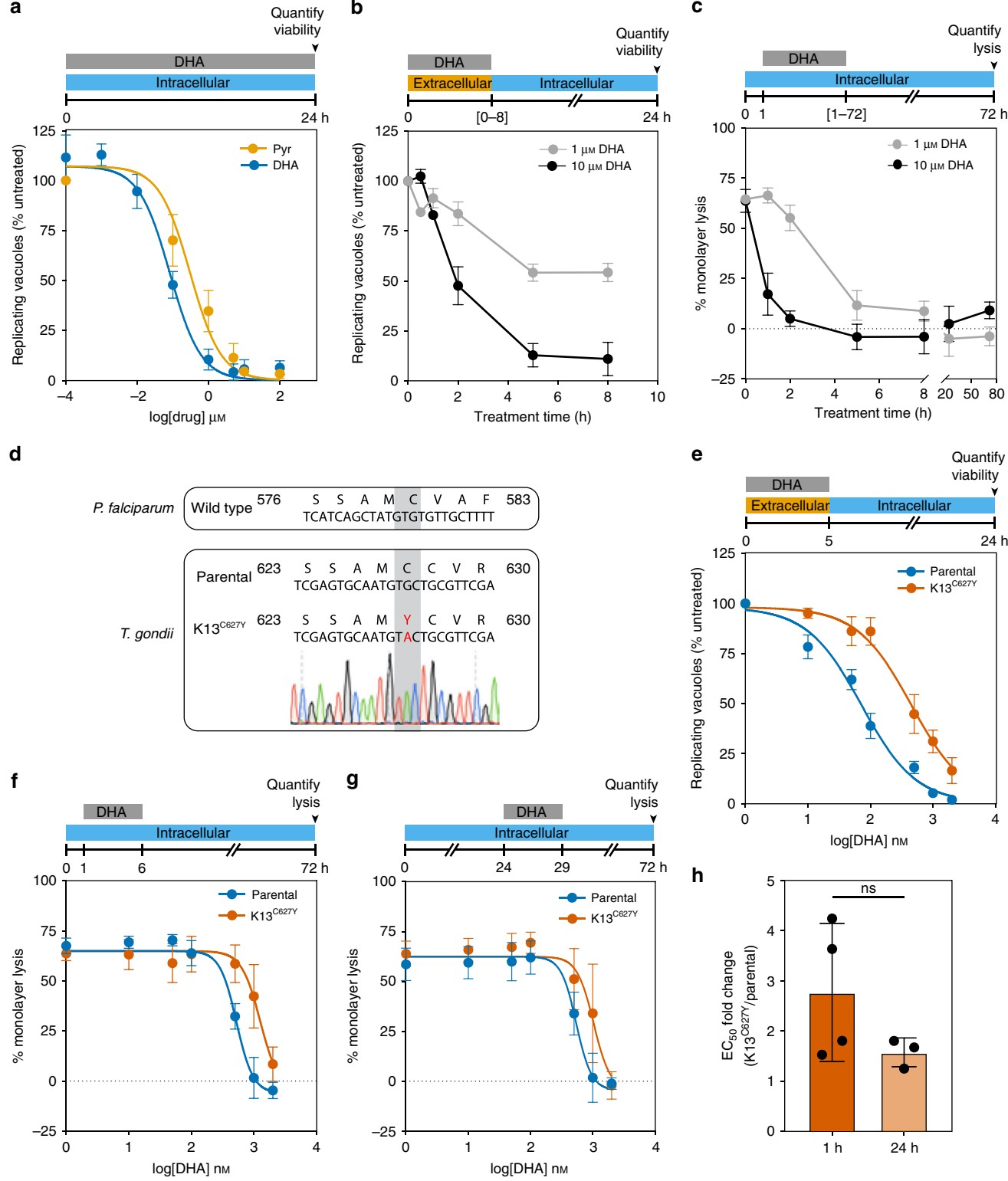

parasites acquire on average a single gRNA that directs Cas9 to create a double-stranded break in the coding sequence of the specified gene[33,34,46]. Insertions and deletions introduced during DNA repair lead to loss-of-function mutations in the targeted genes, and the prevalence of different mutants in the population can be inferred from the relative abundance of gRNAs against each gene. Following generation of the pool of mutants, we propagated this pool in the presence or absence of a sublethal dose (50 nM) of DHA for 7 days (three passages for the untreated

population; Fig. 2a). Guides against a single gene (TGGT1_228110) were substantially depleted under drug treatment compared to the untreated control (Fig. 2b and Supplementary Data 1). TGGT1_228110 encodes a previously unstudied *T. gondii* gene with three to four transmembrane domains, predicted to be dispensable under standard growth conditions[34]. Structural prediction using HHPred[47] indicated that TGGT1_228110 is homologous to TMEM14C ($p = 5 \times 10^{-21}$), a recently identified porphyrin transporter found in mammals[48].

**Fig. 1 DHA kills wild-type *T. gondii* efficiently and its effect is influenced by mutation of K13. a** Treatment with increasing concentrations of dihydroartemisinin (DHA) or pyrimethamine (Pyr) over 24 h resulted in a reduction in parasite viability, measured as the number of vacuoles with two or more parasites normalized to untreated wells. Results are mean ± SEM for $n = 4$ independent experiments. **b** Viability was similarly assessed for extracellular parasites treated with 1 or 10 µM DHA for varying periods of time, then washed and allowed to invade host cells. Viable vacuoles were normalized to untreated parasites kept extracellular for equal periods of time. Results are mean ± SEM for $n = 3$ independent experiments, each performed in technical triplicate. **c** Quantification of host monolayer lysis after treatment with 1 or 10 µM DHA for the indicated time, normalized to uninfected monolayers. Results are mean ± SEM for $n = 4$ independent experiments. **d** Alignment of the mutated region of K13 in *P. falciparum* and *T. gondii* displaying the chromatogram for the K13$^{C627Y}$ *T. gondii* line. **e** Extracellular dose–response curve for parental or K13$^{C627Y}$ parasites treated with DHA for 5 h. Results are mean ± SEM for $n = 7$ or 5 independent experiments with parental or K13$^{C627Y}$ parasites, respectively. **f** Monolayer lysis following infection with parental or K13$^{C627Y}$ parasites for 1 h, prior to a 5 h treatment with varying concentrations of DHA. Results are mean ± SEM for $n = 4$ independent experiments. **g** Monolayer lysis following infection with parental or K13$^{C627Y}$ parasites for 24 h, prior to 5 h treatment with varying concentrations of DHA. Results are mean ± SEM for $n = 4$ or 3 independent experiments with parental or K13$^{C627Y}$ parasites, respectively. **h** Graph summarizing the fold change in DHA EC$_{50}$ between parental and K13$^{C627Y}$ parasites, treated 1 or 24 h after invasion. Results are mean ± SD for $n = 4$ or 3 independent experiments, $p$-value calculated from two-tailed unpaired $t$-test.

Based on this homology, we named the *T. gondii* gene *Tmem14c*. *Tmem14c* is conserved in coccidia but absent from *Plasmodium* spp., suggesting that this gene plays an accessory role in heme synthesis.

To validate the effect of Tmem14c depletion on DHA susceptibility, we replaced the coding region of *Tmem14c* with an mNeonGreen expression cassette (Supplementary Fig. 2). Although Δ*Tmem14c* grew normally under standard conditions, treating mutant parasites with 100 nM DHA for 7 days abolished plaque formation, with minimal effects on the parental line (Fig. 2c). We quantified the change in DHA susceptibility by treating extracellular parasites with varying concentrations of DHA, then measuring their viability. Results showed a modest (twofold) increase in DHA susceptibility for Δ*Tmem14c* parasites compared to the parental line (Fig. 2d and Supplementary Table 1).

We considered that treating intracellular parasites would better resemble the conditions of the CRISPR screens, which maintained DHA pressure throughout the lytic cycle. We therefore assessed the DHA susceptibility of Δ*Tmem14c* parasites by treating with DHA for 5 h immediately after invasion or during replication (Fig. 2e and Supplementary Table 1). In contrast to the parental and K13$^{C627Y}$ parasite lines (Fig. 1h), the susceptibility of Δ*Tmem14c* to DHA was significantly more pronounced during replication, when it was sevenfold greater than that of the parental strain.

To examine the function of Tmem14c, we determined its subcellular localization and the effect of its deletion. A C-terminally Ty-tagged copy of Tmem14c was found to localize to the parasite mitochondrion (Fig. 2f), in agreement with the localization of the mammalian homolog to the inner mitochondrial membrane[48]. We then probed for global metabolic changes in intracellular Δ*Tmem14c* parasites by targeted metabolomics (Fig. 2g and Supplementary Data 2). We observed changes in glycine–serine metabolism, as well as TCA cycle intermediates, which supports the link between Tmem14c and mitochondrial function.

**Genome-wide screens implicate TCA and heme biosynthesis pathways in DHA susceptibility**. We performed three genome-wide CRISPR screens, similar to those described above, to identify genes that when disrupted decrease susceptibility to DHA. Following selection for integration of the gRNAs, parasite populations were treated with DHA or vehicle, then cultured for the time required to passage the untreated population three times (~6 days). The log$_2$-fold change of gRNA abundance between the drug-treated and untreated populations (drug score) was used to rank all protein-coding genes (Fig. 3a and Supplementary Data 1). Two replicates of this screen were performed using 0.5

µM DHA, pooled, and analyzed using Model-based Analysis of Genome-wide CRISPR/Cas9 Knockout (MaGECK)[49]. A third screen performed with 10 µM DHA was analyzed separately. In total, gRNAs against eight genes were significantly enriched in DHA-treated populations in at least two of the three high-dose screens (Supplementary Table 2). A further 65 genes were significantly enriched in individual screens (Supplementary Data 3). The likelihood of identifying a given candidate depends on the gene's contribution to overall fitness, as well as the gene's impact on DHA susceptibility. For every iteration of the screen, the rate at which mutants are lost from the population will fluctuate such that certain fitness-conferring mutants may be completely lost from the population before they have a chance to impact survival under DHA treatment. Even candidates identified in a single screen are significant based on the concordant effect of multiple gRNAs; however, we have the highest confidence in hits obtained from multiple independent screens and focused subsequent analyses on these candidates. Because *K13* mutants are exceedingly difficult to recover by selection with DHA in culture[12] and K13 is necessary for *T. gondii* fitness[34], we did not expect enrichment of gRNAs targeting *K13*, which would predominantly cause loss-of-function mutations.

We performed metabolic pathway analysis on the 73 candidate genes identified by the screens (65 genes identified in single replicates and 8 genes identified in multiple replicates) and found that TCA cycle enzymes were significantly enriched (Bonferroni-adjusted $p$-value $= 1.67 \times 10^{-5}$). The TCA cycle and heme biosynthesis are linked through the production of succinyl-CoA, which is a substrate for δ-aminolevulinic acid synthase (ALAS; Fig. 3b). Two enzymes from the heme biosynthesis pathway were among the candidates: porphobilinogen deaminase (TGGT1_271420) and protoporphyrinogen oxidase (TGGT1_272490). We also identified pyridoxal kinase, which functionalizes the cofactor pyridoxal phosphate. This cofactor is required by several enzymes, including ALAS, which catalyzes the first step of heme biosynthesis[50]. Taken together, these screens support the conclusion that genetically inhibiting heme biosynthesis reduces the DHA susceptibility of *T. gondii*.

**Chemical modulation of heme biosynthesis decreases DHA susceptibility**. We investigated whether inhibitors of the TCA cycle (sodium fluoroacetate, abbreviated as NaFAc)[51] or heme biosynthesis (succinyl acetone, abbreviated as SA)[52] would recapitulate our genetic results. After pretreating extracellular parasites for 2 h with NaFAc or SA, we treated them for a further 5 h with increasing concentrations of DHA before determining parasite viability. Consistent with our genetic results, both NaFAc and SA significantly increased the EC$_{50}$ of DHA compared to untreated parasites (Fig. 3c, d and Supplementary Table 1). To

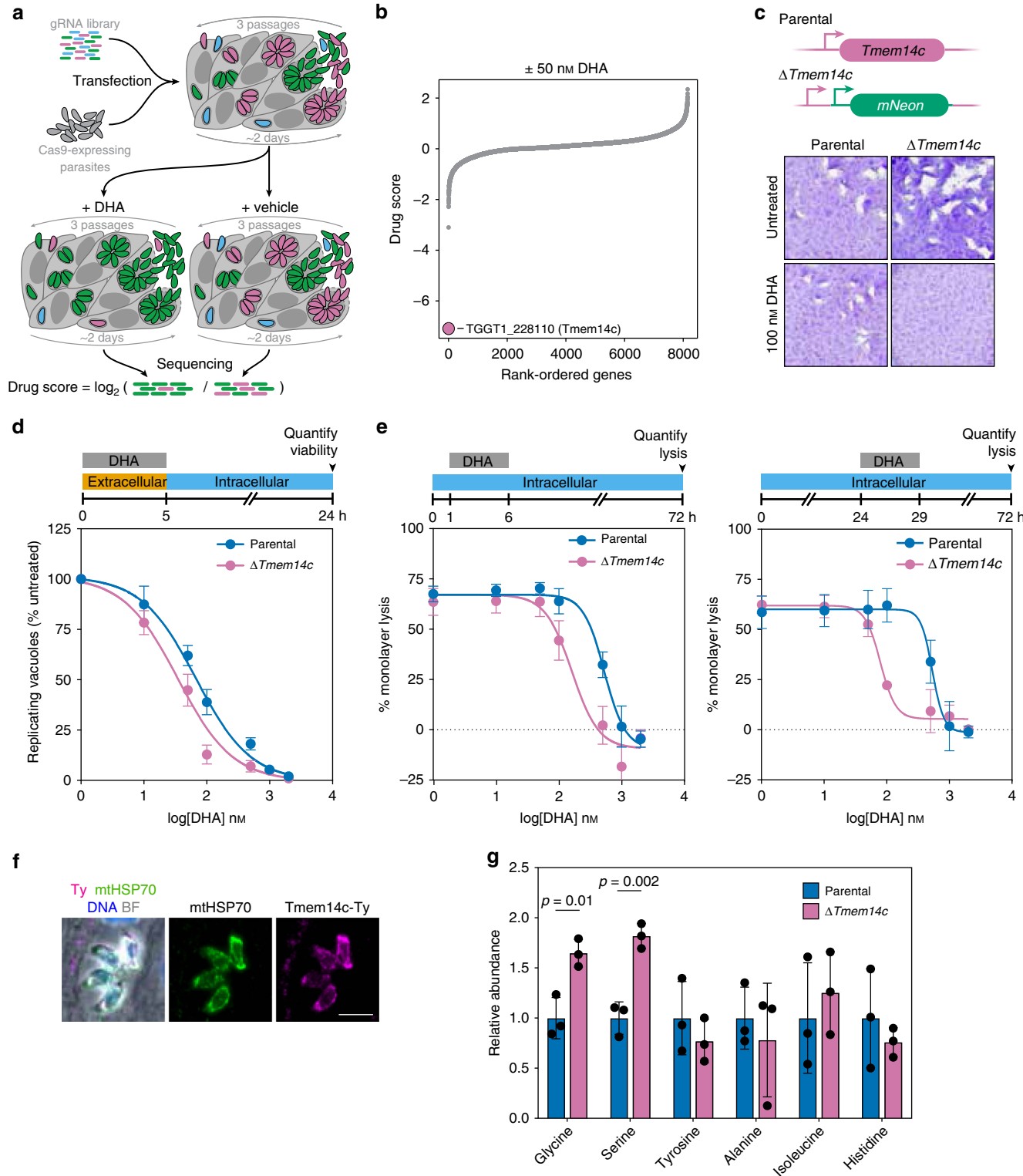

ensure that metabolic modulation alone did not impact DHA susceptibility, we blocked glycolysis using 5 mM 2-deoxyglucose (2-DG)[51,53], as glycolysis is dispensable for TCA activity in the presence of glutamine[54]. Pretreatment with 2-DG did not change the susceptibility of parasites to DHA (Fig. 3e and Supplementary Table 1).

To confirm the effect of the inhibitors on heme production, we quantified total levels of porphyrins—including heme and its precursor protoporphyrin IX (PPIX)—by measuring porphyrin fluorescence[55]. Porphyrin levels were slightly increased by treatment with 2-DG, whereas treatment with NaFAc or SA significantly depressed porphyrin levels (Fig. 3f). We also monitored the effects of 2-DG, NaFAc, and SA by global polar metabolite profiling. Only subtle changes resulted from SA treatment. However, we observed the expected changes in TCA cycle or glycolytic intermediates and neighboring pathways after NaFAc or 2-DG treatment, respectively (Supplementary Fig. 3 and Supplementary Data 4). These results demonstrate that the

**Fig. 2 Genome-wide screen under sublethal DHA concentration reveals that loss of Tmem14c increases DHA susceptibility. a** Screening workflow. The drug score is defined as the $\log_2$-fold change in the relative gRNA abundance between the DHA-treated and vehicle populations, where lower scores are indicative of genes that enhance drug susceptibility when disrupted. **b** Results of a genome-wide CRISPR screen (calculated as described in **a**) comparing treatment with 50 nM DHA (equivalent to $EC_5$) to vehicle-treated parasites. Guides against one gene, Tmem14c, were significantly depleted by the sublethal DHA concentration, but were retained in the vehicle control. **c** $\Delta Tmem14c$ parasites formed plaques normally under standard growth conditions, but their plaquing ability was attenuated in the presence of 100 nM DHA. The parental strain proliferated normally in both conditions. **d** Extracellular $\Delta Tmem14c$ parasites had a decreased $EC_{50}$ compared to the parental line ($p = 0.0003$, extra-sum-of-squares $F$-test). Results are mean ± SEM for $n = 7$ independent experiments. **e** Monolayer lysis following infection with parental or $\Delta Tmem14c$ parasites and treatment with varying concentrations of DHA during hours 1–6 (top panel) or 24–29 (bottom panel) post infection. Results are mean ± SEM for $n = 4$ or 3 independent experiments for parental or $\Delta Tmem14c$ parasites, respectively. **f** Overexpression of Tmem14c-Ty co-localized with mitochondrial marker mtHSP70. Scale bar is 5 μm. **g** Relative abundance of selected amino acids from targeted metabolomics of intracellular parental or $\Delta Tmem14c$ parasites. Results are mean ± SD for $n = 3$ technical replicates, with $p$-values calculated by one-way ANOVA.

TCA cycle is required to maintain porphyrin pools in *T. gondii*, joining the metabolic pathways identified in our screens and confirming that modulation of heme levels can result in decreased DHA susceptibility in apicomplexans.

To test whether increased flux through the heme biosynthesis pathway could hypersensitize parasites to DHA, we supplemented the growth medium with ALA, a precursor that stimulates heme biosynthesis in many organisms[56–58]. While ALA treatment increased total porphyrin levels it did not alter DHA susceptibility, leading us to suspect that this increase in porphyrins was due to a buildup of PPIX, and is not reflective of increased heme levels (Supplementary Fig. 3).

**DegP2 mutants have decreased porphyrin levels and decreased DHA susceptibility.** Having verified the relationship between heme biosynthesis and DHA susceptibility, we turned our attention to TGGT1_290840—the most significant hit in our lethal dose DHA screen lacking a known relationship to heme biosynthesis (Fig. 3a). Previously uncharacterized in *T. gondii*, TGGT1_290840 shares homology with the DegP family of serine proteases. A related *T. gondii* rhoptry protease was recently named DegP[59], so we named TGGT1_290840 DegP2. DegP2 possesses an identifiable catalytic serine (S569), evident in an alignment with the *Arabidopsis thaliana* plastid protease Deg2[60–62].

We generated a panel of strains to study the localization and function of DegP2. These included a DegP2 knockout made by replacing the coding sequence with YFP ($\Delta DegP2$; Fig. 4a). We complemented this knockout with a C-terminally HA-tagged allele expressed from the *TUB1* promoter ($\Delta DegP2/DegP2$-HA). We also endogenously tagged DegP2 with a C-terminal Ty epitope, then mutated DegP2's catalytic serine, creating two strains that we refer to as DegP2-Ty and DegP2$^{S569A}$-Ty, respectively (Fig. 4b). Both the endogenously Ty-tagged and ectopically expressed HA-tagged DegP2 were co-localized with the mitochondrial marker mtHSP70 (Fig. 4b).

Finally, we generated a DegP2 conditional knockdown (cKD) using the U1 system[63]. Using a parental strain expressing a rapamycin-dimerizable Cre recombinase (DiCre), the DegP2 locus is appended with a floxed synthetic 3′ untranslated region (UTR) followed by multiple U1-binding sites. Adding rapamycin to this strain activates Cre, and leads to excision of the 3′-UTR and positioning U1-binding sites directly after the DegP2 coding sequence, altering the stability of the mRNA (Fig. 4c). We used immunofluorescence microscopy (Fig. 4c) and immunoblotting (Fig. 4d) to show that a 2 h pulse of rapamycin leads to robust DegP2 depletion.

Plaque assays showed that both $\Delta DegP2$ and the complemented strain $\Delta DegP2/DegP2$-HA had significant growth defects (Fig. 5a). By contrast, DegP2-Ty and DegP2$^{S569A}$-Ty lines formed plaques similarly to the parental strain (Fig. 5b). This led us to

suspect that $\Delta DegP2$ harbors alterations that extend beyond the activity of the protease. Therefore, we have only attributed to DegP2 those phenotypes that could be complemented in $\Delta DegP2/DegP2$-HA or determined using the DegP2 cKD.

As many of the hits from our lethal dose DHA screen were related to heme biosynthesis—a process that takes place partially in the mitochondrion, where DegP2 is found—we wondered if disrupting DegP2 might also decrease parasite heme levels. By measuring porphyrin fluorescence, we found that $\Delta DegP2$ and DegP2$^{S569A}$-Ty both had significantly reduced porphyrin levels, and porphyrin abundance was partially restored in $\Delta DegP2/DegP2$-HA (Fig. 5c).

To confirm that $\Delta DegP2$'s decreased heme levels correspond to a decrease in DHA susceptibility, we determined the $EC_{50}$ of DHA against the DegP2 knockout and catalytically inactive strains, along with their respective controls (Fig. 5d and Supplementary Fig. 4). As expected, we observed a significantly higher $EC_{50}$ for $\Delta DegP2$ compared to the parental strain (358 nM compared to 76 nM). This effect was largely reversed in $\Delta DegP2/DegP2$-HA (105 nM). Likewise, DegP2$^{S569A}$-Ty had a significantly higher $EC_{50}$ than DegP2-Ty (153 nM compared to 63 nM). When we looked for a correlation between porphyrin levels and DHA $EC_{50}$, we found a strong negative relationship between these strain's porphyrin levels and DHA susceptibility ($r^2 = 0.75$; Fig. 5e).

We confirmed the relationship between DegP2, porphyrin levels, and DHA susceptibility using the DegP2 cKD. Rapamycin-treated DegP2 cKD parasites form plaques similarly to DiCre parasites (Fig. 5f). Using fluorescence to quantify the porphyrin levels in the DegP2 cKD revealed a drop of ~30% in response to rapamycin treatment, similar to the difference observed between $\Delta DegP2$ and its parental strain (Fig. 5g).

To assess the ability of DegP2 cKD parasites to survive DHA treatment, we treated the DegP2 cKD strain with rapamycin, or vehicle, then mixed it in a 1:1 ratio with rapamycin-treated DiCre parasites expressing tdTomato. Rapamycin treatment of the DiCre strain had no effect on its fitness and provided an ideal control for treatment of the cKD. We cultured the resulting populations in the presence or absence of DHA for 8 days and used flow cytometry to analyze the composition of the populations every 2 days (Fig. 5h). Rapamycin-treated DegP2 cKD parasites consistently outcompeted DiCre parasites in the presence of DHA, whereas levels of both strains remained close to 50% under control conditions. Collectively, these results confirm that reducing DegP2 levels reduces porphyrin levels, resulting in decreased susceptibility to DHA.

The decreased porphyrin levels in $\Delta DegP2$ prompted us to examine metabolic changes in other mutants that modulate DHA susceptibility. $\Delta Tmem14c$ and K13$^{C627Y}$ parasites displayed similar levels of porphyrins as their isogenic controls (Supplementary Fig. 2). Analysis of polar metabolites in the K13$^{C627Y}$ strain revealed that the metabolic pathway for alanine, aspartate,

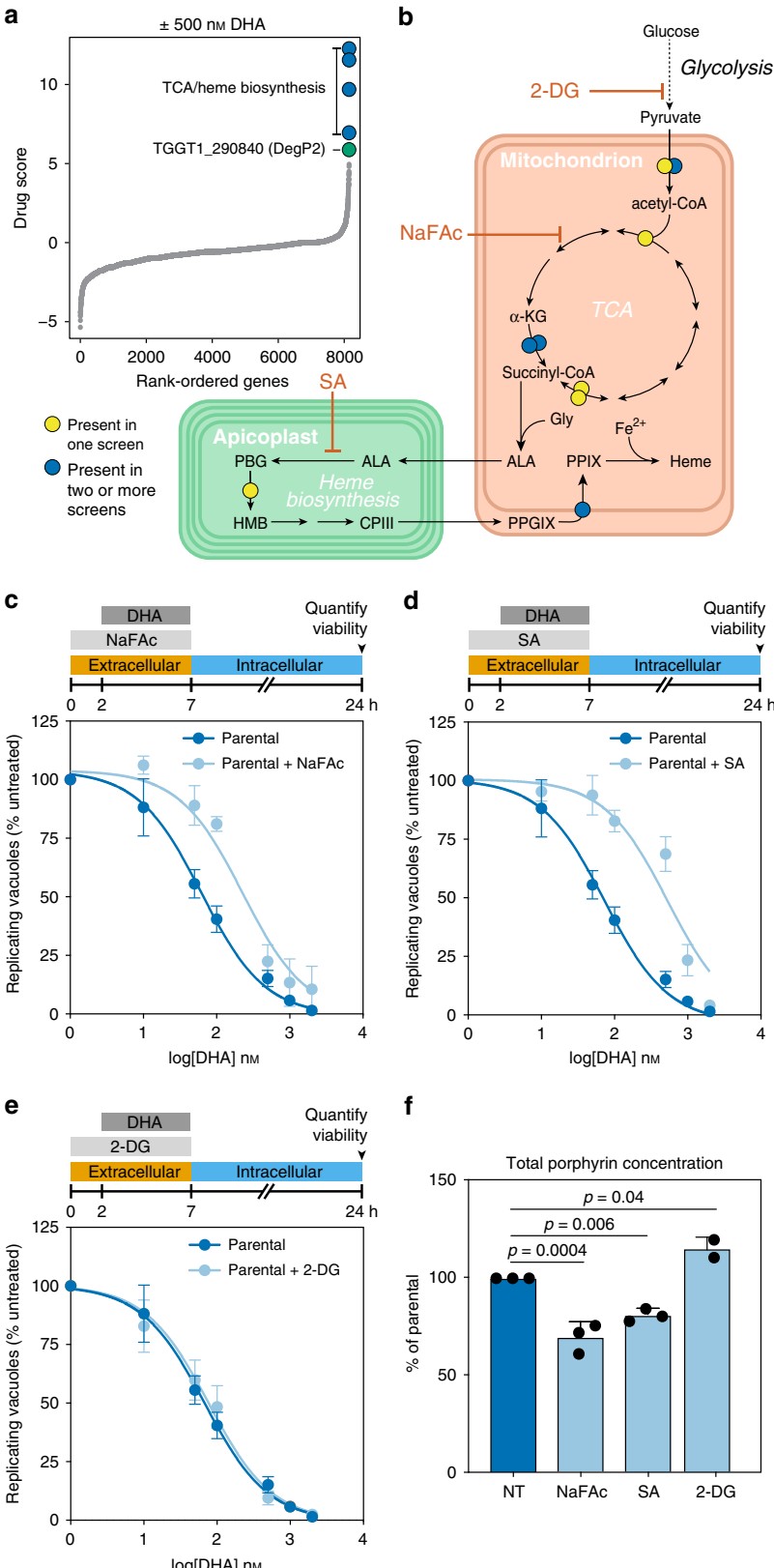

and glutamate production changed significantly (FDR < 0.05). The TCA cycle was also affected (FDR < 0.1), with several intermediates displaying decreased abundance compared to the control (Supplementary Fig. 2 and Supplementary Table 3). Unlike for NaFAc (Supplementary Fig. 3) or ΔDegP2 (discussed below), this effect does not point to a specific block in the TCA cycle, but rather a global dysregulation of the pathway. Our results suggest that in *T. gondii*, ΔTmem14c and K13[C627Y] mutations affect DHA susceptibility independently from the steady-state heme pools. However, we cannot rule out changes in subcellular localization of heme, as may be expected from the predicted role of Tmem14c in porphyrin transport[14,15,48].

**Fig. 3 Genome-wide CRISPR screen under high DHA pressure identifies TCA and heme biosynthetic pathways as determinants of DHA susceptibility.**
**a** Aggregated drug scores from two independent genome-wide CRISPR screens performed with a lethal dose of 500 nM DHA. Drug scores defined as the $\log_2$-fold change in the relative gRNA abundance between the DHA-treated and vehicle populations, where higher scores are indicative of genes that reduce drug susceptibility when disrupted. Genes with consistently high drug scores are highlighted, indicating that their disruption favored parasite survival under otherwise lethal concentrations of DHA. **b** Diagram of key metabolic pathways highlighting genes with high drug scores in the screens from **a** and a third screen performed with 10 μM DHA, analyzed separately. Predicted localization of enzymes within organelles for the TCA cycle and heme biosynthesis pathways. 2-DG 2-deoxyglucose, α-KG α-ketoglutarate, ALA δ-aminolevulinic acid, CPIII coproporphyrinogen III, Gly glycine, HMB hydroxymethylbilane, NaFAc sodium fluoroacetate, PGB porphobilinogen, PPIX protoporphyrin IX, PPGIX protoporphyrinogen IX, SA succinyl acetone. Hits found in two or more replicates (blue) or only one replicate (yellow) are indicated. **c** Pretreatment with 500 mM NaFAc for 2 h significantly increased the $EC_{50}$ of DHA (extra-sum-of-squares $F$-test, $p < 0.0001$). Results are mean ± SEM for $n = 7$ or 4 independent experiments with vehicle or NaFAc treatment, respectively. **d** Pretreatment with 10 mM SA significantly increased the $EC_{50}$ of DHA (extra-sum-of-squares $F$-test, $p < 0.0001$). Results are mean ± SEM for $n = 7$ or 5 independent experiments with vehicle or SA treatment, respectively. **e** Pretreatment with 5 mM 2-DG did not significantly affect the $EC_{50}$ of DHA (extra-sum-of-squares $F$-test, $p = 0.6$). Results are mean ± SEM for $n = 7$ or 3 independent experiments with vehicle or 2-DG treatment, respectively. **f** Porphyrin levels (combination of heme and PPIX) in parasites exposed to various compounds, measured by fluorescence and normalized to untreated parasites (NT). Results are mean ± SD for $n = 3$ independent experiments, each performed in technical duplicate; $p$-values from a one-way ANOVA with Tukey's test.

**DegP2 influences the intersection of the TCA cycle, heme biosynthesis, and the ETC.** We attempted to use immunoprecipitations with either wild type or catalytically dead DegP2 to define DegP2's binding partners, but we were unable to reliably detect interactions. Instead, we turned to thermal proteome profiling (TPP) to identify proteins that change in thermal stability when DegP2 is knocked down[64–67], implying a direct or indirect interaction with DegP2. We collected DegP2 cKD parasites that had been treated with rapamycin or a vehicle control, then heated them to temperatures between 37 °C and 67 °C. After lysis and ultracentrifugation, only non-denatured proteins remained in the supernatant. Using liquid chromatography mass spectrometry (LC–MS), we compared the abundances of non-denatured proteins across the range of temperatures and identified proteins that had altered melting temperatures in the absence of DegP2. We calculated melting temperatures for 1669 proteins across two replicates of the TPP experiment and identified 13 proteins that displayed greater than average changes in both replicates ($p < 0.2$ by $z$-test; Fig. 6a and Supplementary Data 5). Of these 13 proteins, three are mitochondrial[68,69] a NifU domain-containing protein (TGGT1_212930), the ATP synthase γ subunit (TGGT1_231910), and an unannotated protein (TGGT1_226500; Fig. 6b).

Two of these three mitochondrial proteins are related to the ETC. The yeast homolog of TGGT1_212930, Nfu1, transfers nascent iron–sulfur clusters to complex II, as well as to the TCA cycle enzyme aconitase[70]. The γ subunit of the ATP synthase is another critical component of the ETC[53]. These observations motivated us to examine the susceptibility of ΔDegP2 parasites to the complex II inhibitor thenoyltrifluoroacetone (TTFA)[71,72], and the cytochrome $b$ inhibitor atovaquone (ATQ)[73,74]. Parental and ΔDegP2/DegP2-HA parasites showed similar responses to TTFA, while ΔDegP2 parasites were significantly more resistant (Fig. 6c and Supplementary Table 1). By contrast, the $EC_{50}$ of ATQ was unchanged between ΔDegP2 and the parental line (Fig. 6d and Supplementary Table 1). This specific effect of DegP2 on complex II is consistent with the known role of Nfu1 in transferring iron–sulfur clusters to complex II.

To determine whether this apparent alteration in complex II activity resulted in a mitochondrial polarization defect, we loaded ΔDegP2 and its parental strain with MitoTracker, then analyzed their fluorescence using flow cytometry (Supplementary Fig. 4). Both strains displayed similar fluorescence profiles, indicating that ΔDegP2 parasites polarize their mitochondrial membrane to a similar extent as the wild type, despite altered complex II activity.

The TCA cycle is a common thread connecting heme biosynthesis and the ETC. Succinate, a TCA intermediate, is a substrate for both heme biosynthesis and the SDHB subunit of complex II, which converts it to the subsequent TCA intermediate, fumarate. In addition, yeast Nfu1 binds aconitase, which uses an iron–sulfur cluster to catalyze the second step in the TCA cycle[70]. To gain further insight into the mitochondrial dysregulation of ΔDegP2 parasites, we used polar metabolite profiling via LC–MS to compare the levels of TCA cycle intermediates in ΔDegP2 and its parental strain (Fig. 6e and Supplementary Data 6). We observed reduced levels of succinate, in addition to altered levels of pyruvate, glutamine, malate, and GABA in ΔDegP2 parasites. These results show that disrupting DegP2 dysregulates multiple aspects of mitochondrial homeostasis, centering around the intersection between the ETC, the TCA cycle, and heme biosynthesis.

**$P.$ $falciparum$ DegP ($PfDegP$) deletion alters heme concentrations and reduces DHA susceptibility during the erythrocytic cycle.** To investigate the role of the $P.$ $falciparum$ DegP2 ortholog, PfDegP (PF3D7_0807700), in ART susceptibility, we disrupted the gene in the Cam3.II strain expressing wild-type K13 (Fig. 7a, b). Cam3.II $K13^{WT}$ is derived from the Cambodian isolate Cam3.II $K13^{R539T}$, which shows decreased susceptibility to DHA[13]. We compared the susceptibility of Cam3.II $\Delta PfDegP$ to Cam3.II $K13^{WT}$, and found that Cam3.II $\Delta PfDegP$ parasites were less susceptible to DHA in both the ring and schizont stages than Cam3.II $K13^{WT}$ parasites. For comparison, the parental strain, Cam3.II $K13^{R539T}$, displayed low DHA susceptibility in the ring stage, as previously described[13]. Surprisingly, Cam3.II $\Delta PfDegP$ and Cam3.II $K13^{R539T}$ showed similar DHA susceptibility in the schizont stage (Fig. 7c, d).

To test whether the changes in DHA susceptibility were correlated with changes in heme concentrations, we fractionated infected cells and measured free heme, as well as heme associated with hemoglobin and hemozoin[75,76]. Trophozoites from either the Cam3.II $\Delta PfDegP$ and Cam3.II $K13^{R539T}$ strains had significantly lower levels of free heme (Fig. 7e), while heme associated with hemoglobin or hemozoin remained unchanged (Fig. 7f, g). The effect of the K13 mutation on trophozoites points to a link between DHA susceptibility and the availability of free heme within the parasite. Taken together, these data also demonstrate that screens in $T.$ $gondii$ can identify resistance alleles that act through mechanisms conserved across the Apicomplexan phylum.

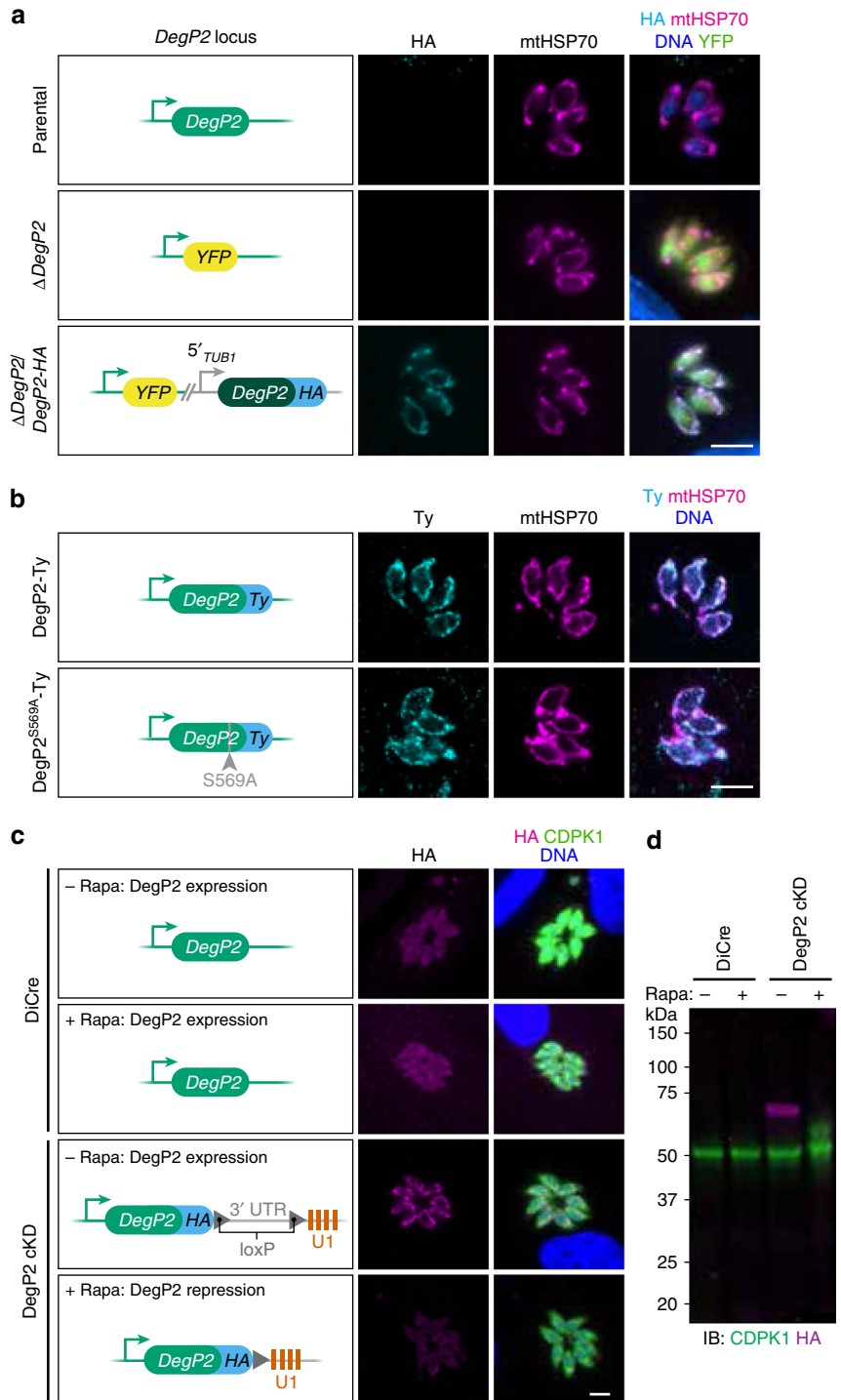

**Fig. 4 Strains generated to study the function and localization of DegP2. a, b** The *DegP2* coding sequence was replaced with *YFP* to generate the *ΔDegP2* line, which was subsequently complemented with an HA-tagged copy of *DegP2* (dark green) under the regulation of the TUB1 promoter (**a**). Separately, the endogenous locus of DegP2 was first tagged with the Ty epitope, and a point mutation was introduced at the catalytic serine (**b**). DegP2-HA, DegP2-Ty, and DegP2S569A-Ty co-localized with the mitochondrial marker mtHSP70. Merged image additionally displays YFP (green) for *ΔDegP2* strains and DNA stain (blue) for the HA-stained samples. Scale bar is 5 μm. **c** A DegP2-inducible mutant constructed using the U1 system. Staining for the HA tag appended to the DegP2 cKD locus showed the expected localization of the protein to the mitochondrion. Individual parasites are visualized by staining for CDPK1. Scale bar is 5 μm. **d** Immunoblotting for DegP2's HA tag showed robust depletion of DegP2 48 h after a 2 h rapamycin treatment. CDPK1 was used as a loading control.

## Discussion

Recent studies have shown that modulating hemoglobin import changes *P. falciparum*'s susceptibility to ART[14,15], and here we extend these results to show that heme biosynthesis plays a role in

*T. gondii*'s susceptibility to this drug. Heme has long been thought to activate ART[4,20,57,77], and our results support the conclusion that heme abundance influences ART susceptibility. These observations help explain why blood-stage *P. falciparum*,

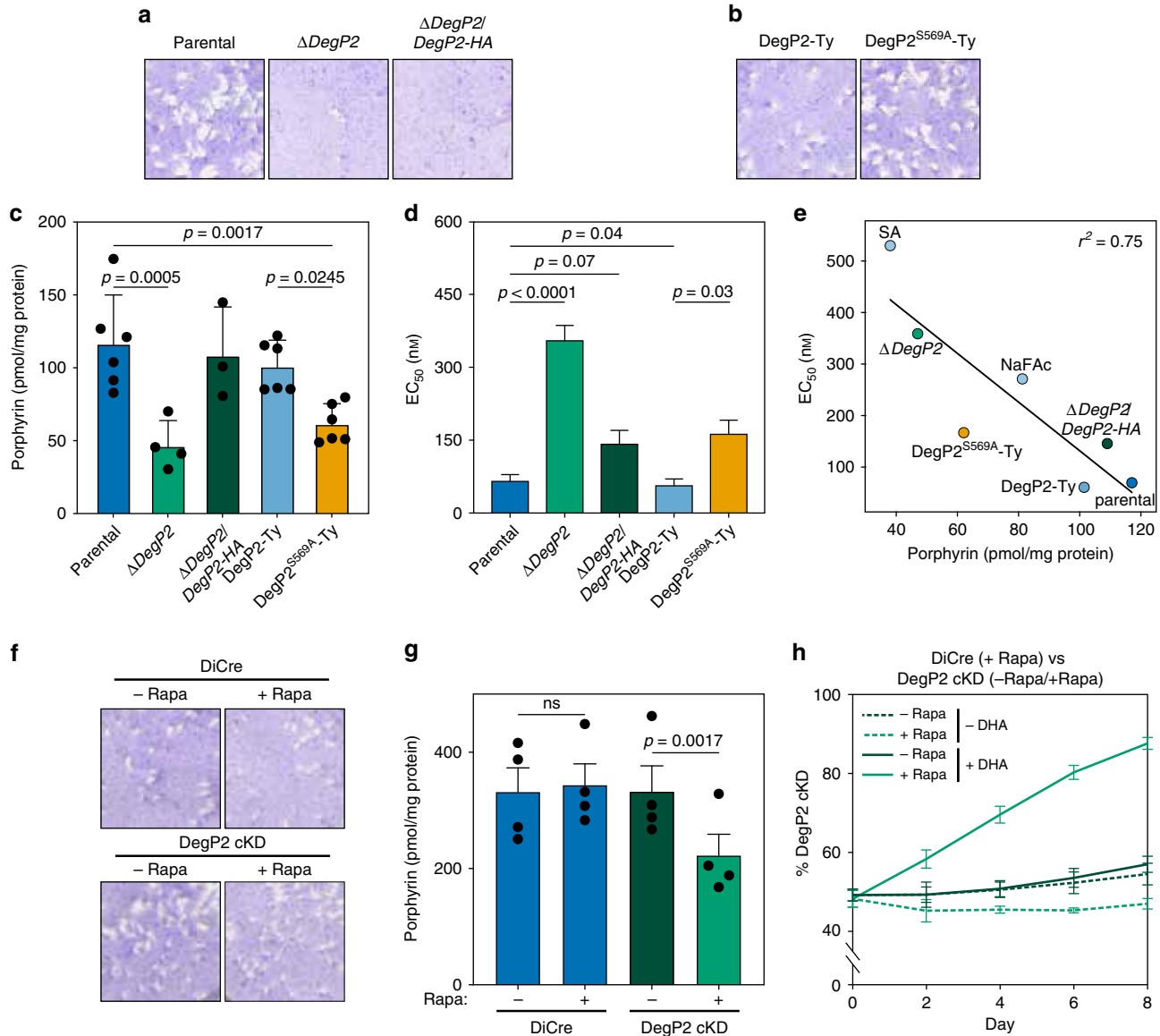

**Fig. 5 Deletion of DegP2 alters porphyrin concentrations and DHA susceptibility. a** Δ*DegP2* and Δ*DegP2/DegP2-HA* parasites formed smaller plaques than their parental strains, indicating a growth defect. **b** Endogenously Ty-tagged DegP2 (DegP2-Ty) parasites and a derived line bearing a mutation in the protease domain (DegP2^S569A-Ty) formed plaques normally. **c** Total porphyrin concentrations (pmol/mg protein) showing reduced porphyrin concentrations for both Δ*DegP2* and DegP2^S569A-Ty parasites. Results are mean ± SD for $n = 3–4$ independent experiments; $p$-values are from one-way ANOVA with Sidak's multiple comparison test. **d** DHA Mean $EC_{50}$ values ± SD were calculated by curve-fitting 4–7 independent experiments together. **e** Porphyrin levels show a negative correlation with DHA $EC_{50}$ ($r^2 = 0.75$) in the strains and conditions tested. **f** DegP2 cKD parasites formed plaques similarly to DiCre, both in the presence or absence of rapamycin. **g** Total porphyrin concentrations for DiCre and DegP2 cKD in the presence or absence of rapamycin. Results are mean ± SEM for $n = 4$ independent experiments; $p$-values are from a two-way ANOVA. **h** DegP2 cKD parasites outcompeted DiCre parasites in the presence of DHA. The fraction of the population composed of mutant parasites was calculated at each time point by flow cytometry. Results are mean ± SEM for $n = 3$ independent experiments.

releasing large amounts of heme from the digestion of hemoglobin, is more susceptible to ART than *T. gondii*[78–80]. Interestingly, *Babesia* spp., which live within erythrocytes but do not take up hemoglobin, have an intermediate susceptibility to ART[81,82], while *Cryptosporidium parvum*—which lacks genes necessary for heme biosynthesis[83,84] shows little response to ART[85].

*T. gondii* and *P. falciparum* differ in their reliance on de novo heme biosynthesis. Inhibiting heme biosynthesis either chemically[52] or genetically[34,86] reduces the fitness of *T. gondii*, highlighting the importance of de novo heme biosynthesis to this parasite. Modulation of heme biosynthesis in cancer cells has similarly been found to alter their susceptibility to ART[57]. By

contrast, heme biosynthesis pathways are dispensable for *P. falciparum* growth during blood stages, although this pathway appears to be necessary during the mosquito stages[87,88]. Although de novo heme synthesis is dispensable for blood-stage *P. falciparum*, the components of this pathway are still expressed, and studies using radiolabeled substrates for heme biosynthesis have shown that the process remains active[89–91]. Our results indicate that there are important parallels between *T. gondii* and *P. falciparum* responses to DHA, despite *T. gondii*'s reduced susceptibility to such compounds. Our results also demonstrate the utility of CRISPR screens in identifying pathways that should be considered when designing ACTs.

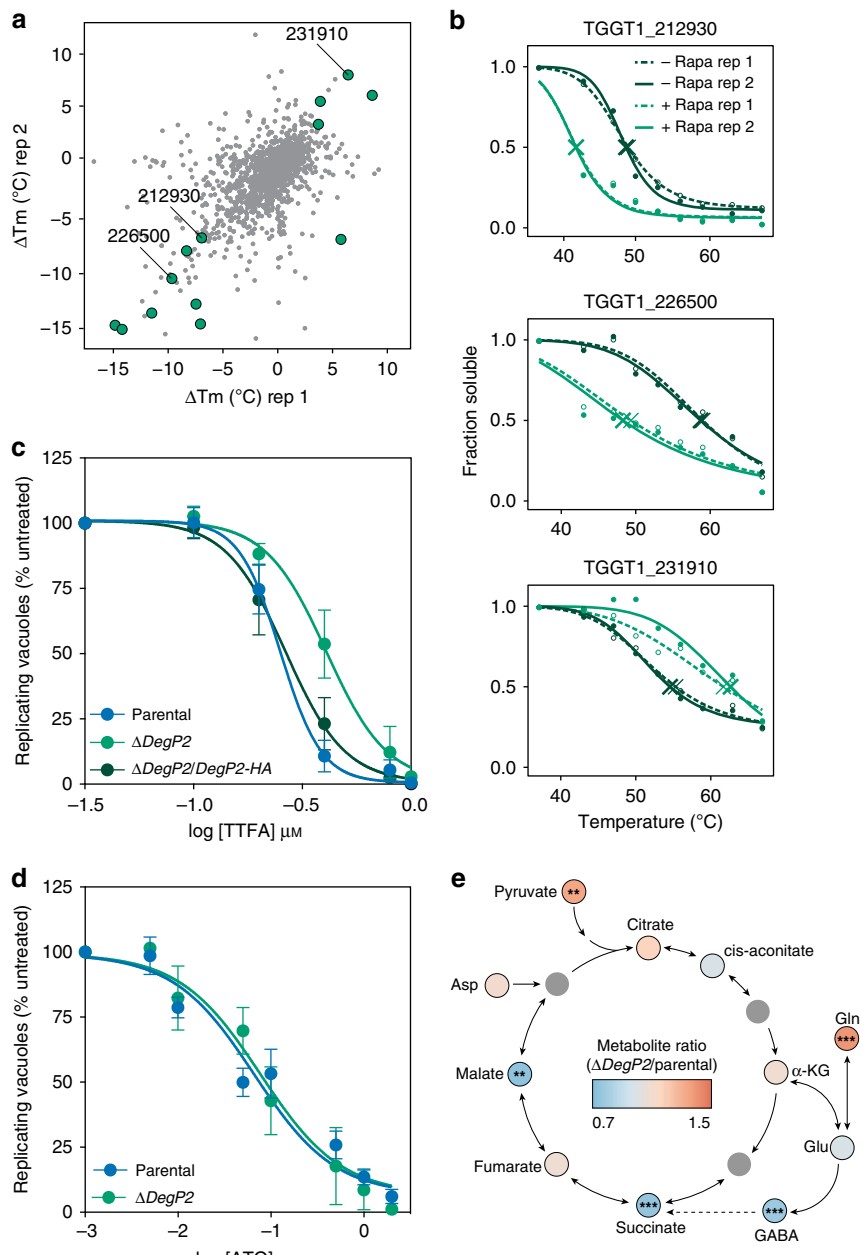

**Fig. 6 DegP2 influences the intersection of heme biosynthesis, the TCA cycle, and the ETC. a** TPP revealed that depleting DegP2 reliably changed the melting temperatures of 13 proteins (highlighted in green, $p < 0.2$ by $z$-test in each of two replicates). Three mitochondrial proteins are indicated by their gene IDs. **b** Melting curves for three mitochondrial proteins identified as hits by TPP. **c** Dose–response curve for parasites treated with the complex II inhibitor TTFA. Results are mean ± SEM for $n = 4$ or 3 independent experiments, for the parental and Δ*DegP2* or Δ*DegP2/DegP2-HA* strains, respectively. Δ*DegP2* is significantly more resistant to TTFA compared to the parental; $p < 0.0001$ from extra-sum-of-squares $F$-test. **d** Dose–response curve for parasites treated for 5 h with increasing concentrations of atovaquone (ATQ). Results are mean ± SEM for $n = 5$ or 3 independent experiments for the parental or Δ*DegP2* strains, respectively. Both strains displayed similar susceptibility to ATQ; $p = 0.59$ from extra-sum-of-squares $F$-test. **e** Summary of changes in metabolites between parental and Δ*DegP2* parasites in the TCA cycle and closely related pathways. Full results can be found in Supplementary Data 6. Asp aspartate, α-KG α-ketoglutarate, Gln glutamine, Glu, glutamate. Asterisks indicate significant change from parental, \**p* < 0.05, \*\**p* < 0.005, \*\*\**p* < 0.0005, by two-way ANOVA.

Mutations in *P. falciparum* K13, particularly K13[C580Y], are major determinants of ART susceptibility in the field. We now know that K13 helps mediate hemoglobin uptake, and that the C580Y mutation reduces DHA susceptibility by reducing protein abundance and limiting flux through this pathway[14,15]. We constructed an analogous mutation in *T. gondii* K13, and found that this mutation also rendered *T. gondii* less susceptible to DHA. *T. gondii* ingests host cytosolic proteins[92], and may take up host

heme or precursors at low levels through endocytosis. Some of *T. gondii*'s heme biosynthesis enzymes can be disrupted—although at a large fitness cost to the parasite[86,93] further suggesting that parasites can scavenge intermediates from the host's heme biosynthetic pathway to sustain viability, but are insufficient for normal growth. It is therefore possible that K13 confers decreased DHA susceptibility through similar mechanisms in *T. gondii* and in *P. falciparum*. Alternatively, recent results show that K13 can

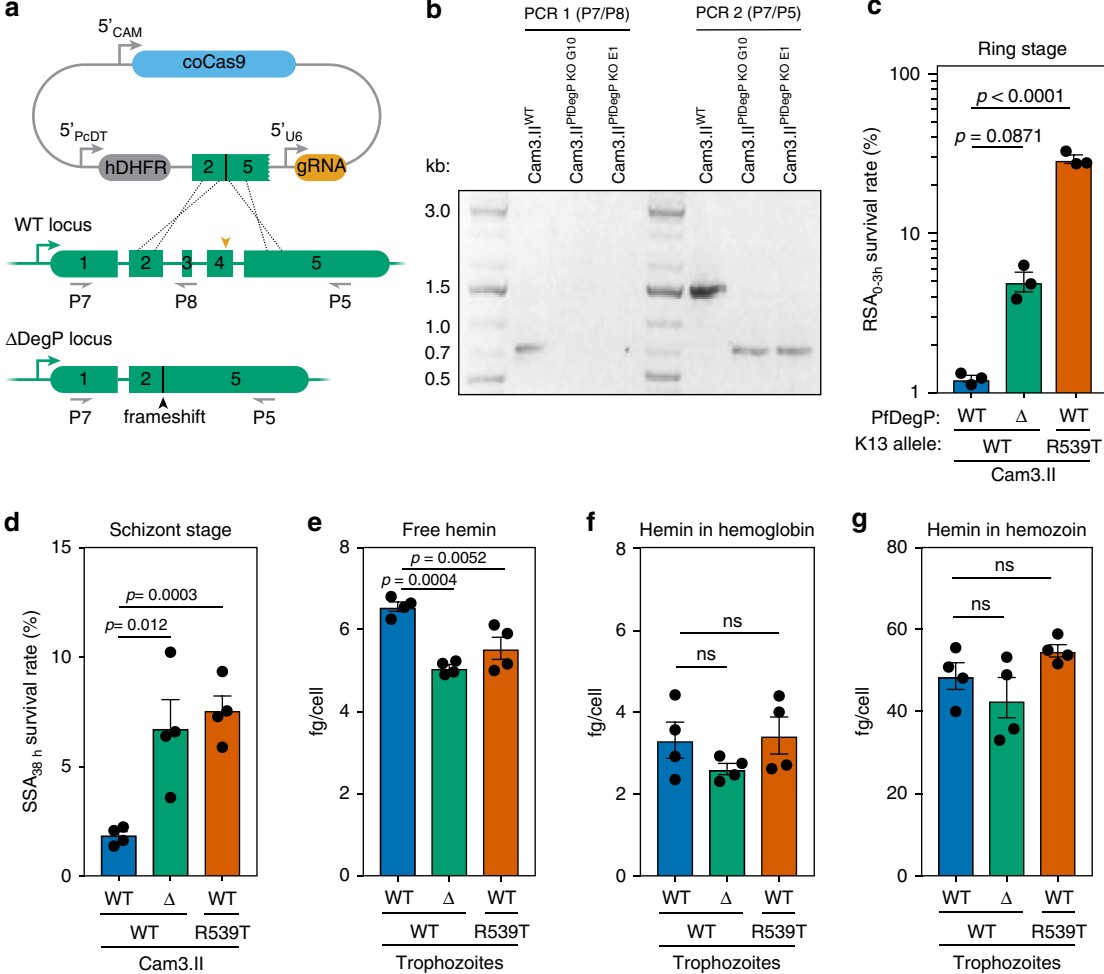

**Fig. 7 Mutating the ortholog of *DegP2* in *P. falciparum* reduces susceptibility to DHA. a** Diagram showing strategy to knockout *PfDegP* in the Cam3.II[WT] background. **b** PCR confirmation of successful KO of *PfDeg2*. PCR 1 (using primers P7 and 8) confirmed the loss of exons 3 and 4 in the two *DegP* KO clones; clone G10 was used for the remainder of this study. PCR 2 (using primers P7 and 5) demonstrated a shorter product upon successful deletion of two exons. **c** Ring-stage survival assay, performed as above, using the Cam3.II[WT], Cam3.II[ΔPfDegP], and the K13 mutant Cam3.II[R539T] lines. Results are mean ± SEM for $n = 3$ independent replicates; *p*-values derive from two-tailed, unpaired *t*-tests. **d** Schizont-stage survival assay following a 4 h pulse of 700 nM DHA. Results are mean ± SEM for $n = 4$ independent replicates; *p*-values derived from two-tailed unpaired *t*-test. **e–g** Heme measured from the free (**e**), hemoglobin-associated (**f**), and hemozoin (**g**) pools from trophozoites for various strains. Heme concentrations were calibrated to a standard curve and normalized to the number of parasites per sample to calculate fg/cell. Results are mean ± SD for $n = 4$ independent replicates; *p*-values from one-way ANOVA.

relocate to the *P. falciparum* mitochondrion upon DHA treatment, indicating that in addition to its role in hemoglobin import, K13 may also influence mitochondrial metabolism[94]. Such alternative functions of K13 are more likely to influence K13's effect on DHA susceptibility in *T. gondii*, given that heme import appears to play a minor role in this parasite compared to biosynthesis.

CRISPR-based screens can identify sensitizing mutations, which may help design combinatorial therapies. Screening for mutants that enhanced susceptibility to a sublethal dose of DHA, we identified an ortholog of TMEM14C, which has been proposed to import porphyrins across the inner mitochondrial membrane of mammalian cells[48]. Although we could not establish a direct role for Tmem14c as a porphyrin transporter and cannot formally exclude alternative roles in mitochondrial metabolism, several lines of evidence lead us to propose that Tmem14c transports heme out of the mitochondrion in *T. gondii*. First, perturbations that decrease heme production decrease DHA susceptibility instead of increasing it, as occurs from Tmem14c disruption. Second, *ΔTmem14c* parasites are most susceptible to

DHA during replication, when increased mitochondrial activity and higher demands on heme production, and export are likely taking place[51]. Third, *ΔTmem14c* parasites have altered levels of glycine, a substrate for heme biosynthesis, which could be explained by negative feedback from excess accumulation of heme in the mitochondrion. ALAS, the enzyme that utilizes glycine to perform the first committed step in heme biosynthesis, has been shown to be regulated by excess heme through a variety of mechanisms[95,96]. Finally, the mechanism and substrate specificity of TMEM14C remain uncharacterized in mammalian cells, leaving open the possibility that TMEM14C might simply mediate passive transport of porphyrins down their concentration gradient. The predicted role of Tmem14c as a porphyrin transporter leads us to propose that changes in heme availability, through its accumulation in the mitochondrion, may influence the susceptibility of parasites to DHA[97,98].

Screening for mutants with decreased DHA susceptibility identified the serine protease DegP2, whose knockout contributed to the survival of *T. gondii* and *P. falciparum* parasites exposed to

DHA. In both species, the activity of DegP2 or PfDegP could be linked to decreased porphyrin concentrations, and in *P. falciparum*, we extended these results to show that levels of free heme were affected by mutating *PfDegP*. *PfDegP* complements its ortholog in *Escherichia coli*, implying broad functional conservation[99].

Using TPP, we uncovered three mitochondrial proteins that may interact with DegP2. One of these hits, TGGT1_212930, contains a NifU domain that in other organisms transfers nascent iron–sulfur clusters to client proteins[100,101]. In yeast, these clients include the TCA cycle enzymes aconitase and SDH2 (SDHB in *T. gondii*)—the latter also part of complex II of the ETC[70]. Bacteria and yeast use chaperones to transfer iron–sulfur clusters from NifU domain-containing proteins to their clients[102–104], raising the possibility that DegP2 fulfills a similar role in *T. gondii*. The complex II inhibitor TTFA[71,72] has an altered EC$_{50}$ when DegP2 is disrupted, which is consistent with DegP2 interacting with the complex II component SDHB, through TGGT1_212930. While this manuscript was in preparation, a study performed using in vitro evolution found that *T. gondii* parasites with point mutations in *DegP2* had decreased ART susceptibility[105]. Interestingly, these researchers also found that such parasites had altered mitochondrial DNA copy number, which in yeast, has been linked to changes in aconitase[106].

In addition to TGGT1_212930, our TPP results revealed that DegP2 influenced the stability of two other mitochondrial proteins: the mitochondrial ATP synthase γ subunit (TGGT1_231910), and a protein annotated as hypothetical (TGGT1_226500). The γ subunit is part of the central stalk around which other components of the ATP synthase complex rotate, to generate ATP[107]. In other systems, Deg-family proteins play roles in the folding, maintenance, and turnover of integral membrane proteins, such as those involved in the ETC[108–110], and a similar phenomenon may explain the apparent association between DegP2 and this ATP synthase subunit. TGGT1_226500 has not been studied extensively, although a global analysis of protein localization predicts it is mitochondrial[69] and perhaps localized to one of the mitochondrial membranes based on the presence of a predicted transmembrane domain[111]. Our studies place DegP2's function at the intersection of the TCA cycle, the ETC, and heme biosynthesis. Despite its pleiotropic effects, parasites can survive without DegP2 under standard growth conditions, albeit with lower porphyrin concentrations that render them less susceptible to DHA.

We wondered whether, analogously to the mutation of K13, disruption of DegP2 would decrease DHA susceptibility in both *T. gondii* and *P. falciparum*. We therefore disrupted the homolog of *DegP2* in *P. falciparum*, *PfDegP*. As in *T. gondii*, these parasites exhibited decreased porphyrin levels and reduced DHA susceptibility. Our results complement recent studies pointing to a role for hemoglobin import in modulating DHA susceptibility[14,15], and suggest that mitochondrial heme dynamics can also have a measurable impact on DHA susceptibility.

Despite critical metabolic differences and over 350 million years of divergent evolution[112], our screens identified multiple genes involved in heme biosynthesis, as critical determinants of DHA susceptibility in *T. gondii*, echoing the results of recent studies that demonstrate that hemoglobin import greatly affects *P. falciparum*'s response to ART[14,15]. Although the two organisms strike drastically different balances between heme import and biosynthesis, the availability of free heme emerges as a limiting determinant of the susceptibility of parasites to ARTs. Our study highlights the power of genome-wide CRISPR screens to identify conserved mechanisms of drug susceptibility and resistance, and points to their potential use in developing novel therapeutics.

## Methods

**T. gondii maintenance and strain construction**. *T. gondii* tachyzoites were grown in human foreskin fibroblasts (HFFs) maintained at 37 °C with 5% CO$_2$ in D3 (Dulbecco's modified Eagle's medium (DMEM, Life Technologies, 11965118) supplemented with 3% heat-inactivated fetal bovine serum and 10 μg/ml gentamicin (Sigma Aldrich, G1272)). When appropriate, chloramphenicol (Sigma Aldrich, C3175) was used at 40 μM and pyrimethamine (Sigma Aldrich, 46706) at 3 μM.

Parasite lines were constructed using CRISPR/Cas9-mediated gene editing as described previously[113] The Cas9-encoding plasmid pU6-Universal is available from Addgene (#52694). Gibson Assembly[114] was used to clone gRNAs into the BsaI sites in pU6-Universal using homology arms of approximately 20 base pairs. Epitope tags and point mutations were introduced by transfecting repair oligos, bearing 40 base pairs of homology with the genome on each end of the mutation or insertion, along with CRISPR machinery. Such repair oligos were often created by heating oligonucleotides encoding complementary single-stranded DNA to 99 °C for 2 min in duplex buffer (100 mM potassium acetate, 30 mM HEPES, pH 7.5), then allowing DNA to cool slowly to room temperature (RT) over the course of several hours.

K13$^{C627Y}$ mutant parasites were constructed by transfecting pU6-Universal encoding the protospacer TTGCTCCTCTCACCACTCCG (oligos P1 and P2 in Supplementary Data 7) into ΔKU80 parasites along with a repair oligo constructed by hybridizing the oligos P3 and P4. Converting the TGC that encodes C627 to TAC eliminated an ItaI site, and restriction digests were used to confirm this mutation. We also introduced a silent CGG to CGA mutation at R667 to eliminate the protospacer adjacent motif and prevent re-cleavage by Cas9 after the initial mutation. Clonal K13$^{C627Y}$ lines were isolated, and the mutation was confirmed by sequencing the locus using primers P5 and P6.

ΔDegP2 parasites were constructed by co-transfecting ΔKU80 parasites with two copies of pU6-Universal encoding gRNAs with the sequences GCAGTCCCCAGCATGGTCGG (P7 and P8 in Supplementary Data 7) and GCGCTCACAAGACCTCGCTGG (P9 and P10) to cleave the 5′ and 3′ ends of the ORF, respectively. These cuts were repaired using a template encoding YFP flanked by 40 nucleotides of homology to the 5′ and 3′-UTRs of *DegP2*, constructed by amplifying the fluorescence marker with P11 and P12. YFP-positive parasites were selected by FACS, and the insertion was confirmed by PCR and Sanger sequencing, using primers P13 and P14.

DegP2-Ty parasites were constructed by transfecting pU6-Universal carrying the gRNA sequence GCGCTCACAAGACCTCGCTGG, together with hybridized P15 and P16, into the ΔKU80 strain. Clonal lines were isolated using limiting dilution, and those carrying the Ty tag were identified using immunofluorescence microscopy[115].

ΔDegP2 was complemented by integrating the plasmid pDegP2-HA into an intergenic region at position 14,87,300 on chromosome VI. pDegP2-HA was constructed by amplifying *DegP2* from cDNA using the primers P17 and P18, and assembling it with a portion of pDsRed[116] that was amplified using the primers P19 and P20. A 200 nucleotide region surrounding the integration locus, with an NheI site at its center, was cloned into the PstI site of the resulting plasmid. We co-transfected NheI-linearized pDegP2-HA with a Cas9-expressing plasmid carrying a gRNA with the sequence GCCGTTCTGTCTCACGATGC[46], then selected for integrants using 40 μM chloramphenicol. Clonal populations carrying DegP2-HA were confirmed using immunofluorescence microscopy with a mouse anti-HA antibody.

DegP2$^{S569A}$-Ty parasites were constructed by transfecting DegP2-Ty parasites with pU6-Universal encoding a protospacer with the sequence GCCATTAATCCTGGCAACAG (oligos P21 and P22), and a repair oligo constructed by annealing P23 and P24. Positive clones were selected based on destruction of a HpyCH4III restriction site, and confirmed by PCR and Sanger sequencing using primers P23 and P24.

DegP2 cKD parasites were constructed by transfecting DiCre parasites[117], with pU6-Universal carrying the gRNA GCGCTCACAAGACCTCGCTGG (P9 and P10) along with a repair template encoding an HA tag followed by the *T. gondii* CDPK3 3′-UTR flanked by two loxP sites, and followed by four U1 recognition sites and a DHFR resistance cassette. This repair template was PCR amplified from a custom-built plasmid using the primers P46 and P47. Following selection with pyrimethamine, clonal lines were selected based on the presence of the HA tag.

ΔTmem14c parasites were constructed by co-transfecting ΔKU80 parasites with two copies of pU6-Universal encoding gRNAs with the sequence TCGGATTGCTATCTGACCAA (oligos P27 and P28) and ACGTCTGATGCCAAGGCGAT (primers P29 and P30), to cleave the 5′ and 3′ ends of the ORF, respectively. These cuts were repaired using a PCR product encoding a *TUB1* promotor, mNeonGreen coding sequence, and the *SAG1* 3′-UTR. This sequence was amplified using the primers P31 and P32. mNeonGreen-positive parasites were selected by FACS, and the insertion was confirmed by PCR and Sanger sequencing, using the primers P33 and P34, P35 and P36, and P37 and P38.

Tmem14c-Ty parasites were created by randomly integrating a plasmid containing Tmem14c-ty into RH parasites. Through PCR and Sanger sequencing of cDNA, the gene model on ToxoDB was found to be incorrect. The stop codon is instead found at the start of exon 2, creating a product of 330 bp. This product was amplified from cDNA using the primers P36 and P39, and assembled into the pSAG1-GCaMP5 (ref. [118]) backbone, in which the chloramphenicol-resistance

cassette was replaced with *DHFR*, in frame with a Ty tag between the *SAG1* promoter and 3'-UTR. Parasites were selected using pyrimethamine, cloned by limiting dilution and positive clones selected by immunofluorescence microscopy for the Ty tag.

DiCre/tdTomato parasites were constructed by integrating a tdTomato expression cassette into an intergenic region at position 1487300 on chromosome VI. DiCre parasites[119] were co-transfected with a Cas9-expressing plasmid carrying a gRNA with the sequence GCCGTTCTGTCTCACGATGC[46] and a repair template encoding a *TUB1* promotor, tdTomato coding sequence, and the *DHFR* 3'-UTR, which was amplified from a custom-built plasmid using the primers P44 and P45. tdTomato-positive parasites were selected by FACS, and a clonal population was isolated using limiting dilution and confirmed by flow cytometry.

**Pooled genome-wide screens.** Genome-wide CRISPR screens were performed based on the previously published method[33]. Briefly, 500 μg of gRNA library linearized with AseI was transfected into ~4 × 10^8 Cas9-expressing parasites[46] divided between ten individual cuvettes. Parasites were allowed to infect 10 × 15 cm^2 plates of confluent HFFs, and the medium changed 24 h post infection to contain pyrimethamine and 10 μg/ml DNaseI. Parasites were passaged upon lysis (between 48–72 h post transfection) and selection continued for three passages. At this point, parasites with integrated gRNA plasmids were split into two pools and either treated with indicated concentrations of DHA or left untreated. Untreated parasites were passaged onto fresh HFFs every 2 days, while the medium on treated parasites was changed for fresh DHA at the same time point. After a further three passages of the untreated population, parasites were collected, counted, and gDNA was extracted, using a DNeasy blood and tissue kit (Qiagen). When parasites were not limiting (e.g., in the untreated sample) DNA was extracted from 1 × 10^8 parasites. When DHA was used for positive selection, DNA was extracted from all recovered parasites. Integrated gRNA constructs were amplified from 500 ng of extracted DNA, using the primers P40 and P41 in two independent reactions and pooled. The resulting libraries were sequenced on a HiSeq 2500 (Illumina) with single-end reads, using primers P42 and P43.

**CRISPR screen data analysis.** Sequencing reads were aligned to the gRNA library. The abundance of each gRNA was calculated and normalized to the total number of aligned reads. To determine the fitness effect of each guide under DHA across biological replicates, we made use of the MAGeCK algorithm[49]. Hits were selected based on FDR of <0.1 in at least one biological replicate. Metabolic pathways analysis on hits was performed using ToxoDB[120] against KEGG and MetaCyc pathways, and the Bonferroni-adjusted *p*-value reported.

**Compounds used with *T. gondii*.** DHA (VWR, TCD3793) was prepared at 10 mM in DMSO as single-use aliquots and used at the indicated concentration. Hemin (Sigma Aldrich, H9039) was prepared at 10 mM in 0.5 M NaOH. 2-DG (Sigma Aldrich, D6134) was prepared fresh at 10 mM in D3 and used at a final concentration of 5 mM. SA (Sigma Aldrich, D1415) was prepared in PBS as 200 mM stocks and used at a final concentration of 10 mM. ALA (Sigma Aldrich, A3785) was prepared in PBS at 200 mM and used at a final concentration of 200 μM. NaFAc (Fisher Scientific, ICN201080) was prepared at 1 M in water and used at a final concentration 500 μM. ATQ (Sigma Aldrich, A7986) was prepared at 27 mM in DMSO and used at the indicated concentration. TTFA (Sigma Aldrich, T27006) was prepared as 10 mM stocks in DMSO and used at the indicated concentrations. MitoTracker Deep Red FM (Life Technologies, M22426) was used at a final concentration of 50 nM. Oligomycin (EMD Millipore, 495455) was used at a final concentration of 20 μM. Pyrimethamine (Sigma Aldrich, 46706) was prepared in ethanol at 10 mM. ADP (A2754-100MG) was prepared at 100 mM in water. FCCP (Sigma Aldrich C2920) was prepared at 50 mM in DMSO.

**Lytic assay.** A total of 50,000 parasites per well were spun down onto HFFs grown in 96-well plates. After 1 or 24 h, the medium was removed and replaced with medium containing vehicle or drug, and parasites were incubated for the indicated time before the wells were washed, the medium replaced, and the monolayers incubated until 72 h.p.i. Monolayers were rinsed with PBS, fixed in 95% ethanol or 100% ice-cold methanol for 10 min, and stained with crystal violet (Sigma, C6158) stain (2% crystal violet, 0.8% ammonium oxalate, and 20% ethanol) for 5 min before washing excess dye with water and allowing to dry. Absorbance of wells was read at 570 nm and normalized to drug treated, uninfected wells. Each experiment was performed with three technical replicates, and results show the mean of at least four independent experiments.

**Pulsed treatment assay.** Extracellular parasites were incubated in a total volume of 100 μl D3 with DHA for the indicated time, spun down, and resuspended in 600 μl of pre-warmed D3 before aliquoting 200 μl of parasites to HFF monolayers on a clear-bottomed plate and incubating for 24 h. Wells were then fixed in 4% formaldehyde for 10 min and stained using rabbit anti-PCNA, and detected using anti-rabbit Alexa 549. Images were acquired using a Cytation 3 high content imager, and the number of vacuoles and parasite nuclei per vacuole were quantified using an automated ImageJ macro. Vacuoles of two or more parasites were considered to be alive, and the number of such vacuoles in drug-treated wells was

normalized to the number in untreated wells to quantify the lethality of drug treatment.

**Porphyrin assay.** Based on published methods[55], ~5 × 10^7 freshly egressed parasites were passed through a 3 μm filter, washed once in 10 ml of PBS, then pelleted and resuspended in 50 μl of $H_2O$, and flash frozen in liquid nitrogen. A total of 20 μl of parasite lysate was then mixed with 200 μl of pre-warmed 1.5 M oxalic acid (Sigma Aldrich, 75688) and incubated in a thermocycler at 99°C for 30 min. The total volume was then transferred to a clear-bottomed plate and fluorescence was detected at 662 nm following excitation at 400 nm. Porphyrin levels were calculated from a standard curve prepared as above using hemin. Total porphyrin was normalized to protein concentration, as calculated by Bradford assay (BioRad, 5000201). Experiments were performed with two technical replicates, and results are representative of at least three independent experiments.

**Immunofluorescence microscopy.** Infected cells were fixed in 4% formaldehyde for 10 min, then permeabilized with 0.25% Triton X-100 for 8 min. Staining was performed with mouse anti-Ty[115], rabbit anti-mtHSP70 (ref. [121]), and mouse anti-HA (1:1000; Biolegends 901513) and detected with Alexa-Fluor-labeled secondary antibodies (1:1,000; Thermo Fisher). Nuclei were stained with Hoechst 33258 (Santa Cruz, SC-394039) and coverslips were mounted in Prolong Diamond (Thermo Fisher, P36961). Images were acquired using an Eclipse Ti epifluorescence microscope (Nikon) using the NIS elements imaging software. FIJI[122] was used for image analysis and Adobe Photoshop for image processing.

**Plaque formation.** A total of 500 parasites per well were used to analyze the effect of drug treatment or gene deletion over the course of 8 days. The monolayers were then rinsed with PBS, fixed in 95% ethanol or 100% ice-cold methanol for 10 min, and stained with crystal violet (2% crystal violet, 0.8% ammonium oxalate, and 20% ethanol) for 5 min.

**MitoTracker analysis of Δ*DegP2*.** Parasites were suspended at 1 × 10^7 parasites per ml in D3 with or without 30 μM oligomycin. After 15 min at 37 °C, the samples were mixed 2:1 with MitoTracker (Thermo Fisher Scientific, M7512) in D3, bringing the final concentration of oligomycin to 20 μM and MitoTracker to 12.5 nM. Parasites were incubated at 37 °C for an additional 30 min, then pelleted, and resuspended in Ringer's solution (115 mM NaCl, 3 mM KCl, 2 mM $CaCl_2$, 1 mM $MgCl_2$, 3 mM $NaH_2PO_4$, 10 mM HEPES, 10 mM glucose, and 1% FBS), before analysis by flow cytometry.

**Extraction of polar metabolites for LC–MS analysis.** Parasites were harvested from HFFs when approximately half the vacuoles were lysed and prepared as for transcriptomics. Parasite pellets were quenched in 300 μl ice-cold 75% ACN supplemented with 0.5 μM isotopically labeled amino acids (Cambridge Isotopes, MSK-A2-1.2), and kept at −20 °C. To extract polar metabolites, samples were sonicated for ten cycles in a Bioruptor (Diagenode) at 4 °C with 30 s on and 30 s off, then incubated at 4 °C for 10 min. After a 10 min, 10,000 r.p.m. spin in a tabletop centrifuge (Eppendorf), the supernatant was collected and the pellet was washed with 100 μl 75% ACN and spun again. The supernatants from the two spins were combined. Extracted metabolites were dried in a CentriVap concentrator equipped with a cold trap (Labconco, Kansas City, MO) and reconstituted in an appropriate volume of 75% ACN (30–100 μl).

**LC–MS targeted analysis for polar metabolites.** Polar samples were treated for small-molecule LC–MS as previously described[123]. MS data acquisition on a Q Exactive benchtop orbitrap mass spectrometer equipped with an Ion Max source and a HESI II probe, which was coupled to a Dionex UltiMate 3000 UPLC system (Thermo Fisher Scientific, San Jose, CA) and was performed in a range of $m/z = 70$–1000, with the resolution set at 70,000, the AGC target at 1 × 10^6, and the maximum injection time (Max IT) at 20 ms. For tSIM scans, the resolution was set at 70,000, the AGC target was 1 × 10^5, and the max IT was 250 ms. Relative quantitation of polar metabolites was performed with XCalibur QuanBrowser 2.2 (Thermo Fisher Scientific) and TraceFinder software (Thermo Fisher Scientific, Waltham, MA), using a 5 p.p.m. mass tolerance and referencing an in-house library of chemical standards. Pooled samples and fractional dilutions were prepared as quality controls and only those metabolites were taken for further analysis, for which the correlation between the dilution factor and the peak area read was >0.95 (high-confidence metabolites). Normalization for relative parasite amounts was based on the total integrated peak area values of high-confidence metabolites within an experimental batch after normalizing to the averaged factor from all mean-centered areas of the isotopically labeled amino acids internal standards. It is of note that normalizing to protein content by Bradford analysis or the sum of all area reads from high-confidence metabolites lead to similar results. The data were further Pareto transformed for MetaboAnalyst-based statistical or pathway analysis[124].

**Immunoblotting.** Parasites were lysed in 20 mM HEPES, 137 mM NaCl, 10 mM $MgCl_2$, 1% Triton X-100 supplemented with Halt protease inhibitors (Thermo

Fisher 78430). Proteins were separated by SDS–PAGE and transferred to nitro-cellulose membranes. CDPK1 was probed using a custom-made antibody[125] (1:20,000) and HA using mouse anti-HA (1:5000; Biolegends 901513).

**Competition assays.** DiCre/tdTomato and DegP2 cKD parasites were subjected to a 2-h treatment with 50 nM rapamycin or vehicle control 3 days prior to beginning competition assays. Rapamycin-treated DiCre/tdTomato parasites were then mixed with either treated or untreated DegP2 cKD parasites in a 1:1 ratio, and cultured in the presence of 0.5 μM DHA or vehicle control. The population compositions were monitored every 2 days using a MACSQuant flow cytometer.

**TPP sample preparation.** For the purpose of processing biological replicates, parasites were grown in heavy (Lys8/Arg10) and light SILAC media for three passages prior to harvest. Between the second and third passage, intracellular parasites were treated with 50 nM rapamycin or vehicle control for 2 h. Two 15-cm dishes per treatment were harvested and passed through 0.2 μm filters (EMD Millipore). The parasites were spun at $1000 \times g$ for 10 min at RT, were washed in 1 ml TPP buffer (142 mM KCl, 1 mM $MgCl_2$, 5.6 mM glucose, and 25 mM HEPES), and were spun again at $1000 \times g$ for 10 min at RT. The parasite pellet was resuspended in 1.2 ml TPP buffer, and 100 μl was aliquoted into PCR-strip tubes. Parasites were heated at 37, 43, 47, 50, 53, 56, 59, 63, and 67 °C for 3 min using the gradient function on two 48-well thermal cyclers (BioRad). The tubes were placed on ice, and the parasites were combined with 20 μl of 6× lysis buffer to a final concentration of 1% IGEPAL CA-680, 1× Halt protease and phosphatase inhibitors, and 250 U/ml benzonase in TPP buffer. Lysates were spun at $100,000 \times g$ for 20 min in a TLA-100 rotor with a Beckman Ultra MAX ultracentrifuge. The concentration of soluble protein in the supernatant in the 37 °C samples was quantified with a DC Protein Assay (BioRad) by comparison to a standard curve of bovine serum albumin.

**Protein digestion and peptide labeling.** A volume corresponding to 50 μg of protein from the 37 °C heavy and light SILAC sample was combined, for 100 μg total. Equivalent volumes from the remaining temperatures were similarly pooled. Samples were reduced with 5 mM TCEP for 10 min at 55 °C, and were alkylated with 15 mM MMTS for 10 min at RT. The samples were desalted and cleared of detergent using the SP3 protocol[126]. Proteins were digested at a 1:50 trypsin:protein ratio in 50 mM HEPES pH 8.5 overnight at 37 °C. Peptides were eluted and quantified using the Pierce Fluorometric Peptide Assay, and 50 μg of peptides from the 37 °C sample and the equivalent volume from the remaining temperatures were labeled with TMT10plex at a 1:2 peptide:tag ratio according to manufacturer's instructions (Thermo Fisher Scientific). Labeled peptides from one treatment condition were pooled and desalted with a SepPak Light (Waters), eluted in 40% acetonitrile, 0.1% acetic acid, and lyophilized. Peptides were fractionated offline via reversed-phase high-performance liquid chromatography using Shimadzu LC-20AD pumps and a 10 cm × 2.1 mm column packed with 2.6 μm Aeris PEPTIDE XB-C18 media (Phenomenex). The gradient was isocratic 1% A buffer (20 mM ammonium formate, pH 10 in water) for 1 min at 150 μl/min with increasing B buffer (100% acetonitrile) concentrations to 16.7% B at 20.5 min, 30% B at 31 min, and 45% B at 36 min. Fractions were collected with a FRC-10A fraction collector, and 15 samples were lyophilized for analysis.

**LC–MS/MS data acquisition.** Each fraction was resuspended in 0.1% formic acid and analyzed on an Orbitrap Q Exactive HF-X mass spectrometer in positive ion mode connected to an EASY-nLC chromatography system, using 0.1% formic acid as solvent A and 80% acetonitrile, 0.1% formic acid as solvent B (Thermo Fisher Scientific). Peptides were separated at 3 μl/min on a gradient of 6–21% B for 41 min, 21–36% B for 20 min, 36–50% B for 10 min, and 50–100% B over 15 min. Full-scan spectra were acquired in profile mode with a scan range of 375–1400 $m/z$, resolution of 120,000, maximum fill time of 50 ms, and AGC target of 3E6 with a 15-s dynamic exclusion window. Precursors were isolated with 0.8 $m/z$ window and fragmented with a NCE of 32. The top 20 MS2 spectra were acquired over a scan range of 350–1500 $m/z$ with a resolution of 45,000, AGC target of 1E5, maximum fill time of 120 ms, and first fixed mass of 100 $m/z$. Raw data files have been deposited to the ProteomeXchange Consortium via the PRIDE partner repository[127,128] with the dataset identifier PXD019917.

**Protein MS data analysis.** Peak lists and protein IDs were generated in Proteome Discoverer 2.2 using Sequest HT (Thermo Fisher Scientific) and the ToxoDB-45 protein database (ToxoDB.org). The search included the following posttransla-tional modifications for light samples: dynamic phospho/+79.966 Da (S, T, Y), dynamic oxidation/+15.995 Da (M), dynamic acetyl/+42.011 Da (N-terminus), dynamic TMT6plex/+229.163 Da (any N-terminus), dynamic TMT6plex/+229.163 Da (K), and static methylthio/+45.988 Da (C); and the following post-translational modifications for heavy samples: dynamic phospho/+79.966 Da (S, T, Y), dynamic oxidation/+15.995 Da (M), dynamic acetyl/+42.011 Da (N-terminus), dynamic TMT6plex/+229.163 Da (any N-terminus), dynamic Lys8-TMT6plex/+237.177 Da (N-terminus), static Lys8-TMT6plex/+237.177 Da (K), static label:13 C(6)15 N(4)/+10.008 Da (R), and static methylthio/+45.988 Da (C). Reporter ion intensities and ratios were quantified for unique peptides with a strict

1% FDR, co-isolation threshold of 50%, and S/N of 5. Protein abundance relative to the sample heated to 37 °C were used as input to the TPP R package[66]. Input and output data are provided in Supplementary Data 5.

***P. falciparum* maintenance and strain construction.** *P. falciparum* asexual blood-stage parasites were cultured in human erythrocytes (3% hematocrit) and RPMI-1640 medium supplemented with 2 mM L-glutamine, 50 mg/l hypoxanthine, 25 mM HEPES, 0.225% $NaHCO_3$, 10 mg/l gentamicin, and 0.5% (w/v) Albumax II (Invitrogen). Parasites were maintained at 37 °C in 5% $O_2$, 5% $CO_2$, and 90% $N_2$. Cultures were stained with Giemsa, monitored by blood smears fixed in methanol, and viewed by light microscopy.

Transfections were performed by electroporating ring-stage parasites at 5–10% parasitemia with 50 μg of purified circular plasmid DNA resuspended in Cytomix[129]. One day after electroporation, parasites were exposed to 2.5 nM WR99210 for 6 days to select for transformed parasites. Parasite cultures were monitored by microscopy for up to 6 weeks post electroporation, and recrudescent parasites were screened for editing by PCR. Positively edited bulk cultures were cloned by limiting dilution in 96-well plates, and flow cytometry was used to screen for positive parasites after 17–20 days. Parasites were stained with 1× SYBR Green (ThermoFisher) and 100 nM MitoTracker Deep Red (Invitrogen), and detected using an Accuri C6 flow cytometer (Becton Dickinson[130]).

To generate the knockout construct for the serine protease *Pf*DegP (PF3D7_0807700) a donor sequence (834 bp total) was synthesized (Genewiz) fusing homology regions located in exon 2 (HR1) and exon 5 (HR2), with the addition of one nucleotide to introduce a frameshift in exon 5. The donor fragment was cloned into the pDC2 plasmid[131] by amplifying with primers p15 and p16 (Supplementary Data 7) using the restriction sites EcoRI and AatII. The pDC2 plasmid[132] contains a codon-optimized Cas9 sequence under the regulatory control of the 5′ calmodulin (PF3D7_1434200) promoter and the 3′ *hsp86* (PF3D7_0708500) terminator, as well as the human *DHFR* (*hDHFR*) selectable marker that mediates resistance to the antiplasmodial agent WR99210 (Jacobus Pharmaceuticals). Two gRNA sequences (Supplementary Data 7) located in exon 4 were selected using chopchop[133] and cloned into pDC2 using an In-Fusion cloning kit (Takara Bio USA, Inc.). The final constructs pDC2-coCas9-*PfDegPKO-hDHFR* were transfected into Cam3.II<sup>WT</sup> parasites as described above. Primers for diagnostic PCRs, as well as sequencing primers are shown in Fig. 6b and listed in Supplementary Data 7.

**Schizont-stage and ring-stage survival assays ($SSA_{38 h}$ and $RSA_{0-3 h}$).** These assays were carried out as previously described[43], with minor modifications. In brief, parasite cultures were synchronized one to two times using 5% sorbitol (Fisher). Drug was maintained throughout the assay until assessment of parasite growth. Synchronized schizonts were incubated in RPMI-1640 containing 15 units/ml sodium heparin for 15 min at 37 °C to disrupt agglutinated erythrocytes, then concentrated over a gradient of 75% Percoll (Fisher) and washed once in RPMI-1640. For the schizont-stage survival assay the concentrated schizonts were counted and seeded at 0.3% parasitemia followed by a 4 h exposure to 700 nM DHA or 0.1% DMSO (vehicle control). For the ring-stage survival assay purified schizonts were incubated for 3 h with fresh red blood cells, to allow time for merozoite invasion. Cultures were subjected again to sorbitol treatment to eliminate remaining schizonts. A total of 0–3 h post invasion rings were adjusted to 1% parasitemia and 2% hematocrit, and exposed to 700 nM DHA or 0.1% DMSO (vehicle control) for 4 h. Cells were washed to remove drug and returned to standard culture conditions for an additional 66 h. Parasite growth in each well was assessed using flow cytometry. Between 60,000 and 100,000 events were captured for each well. After 72 h, cultures generally expanded to 3–5% parasitemia in DMSO-treated controls. Percent survival of DHA-treated parasites was calculated relative to the corresponding DMSO-treated control.

**Cellular heme fractionation assay.** Heme fractionation was performed based on previously described methods[75,76]. Briefly, sorbitol-synchronized early ring-stage parasites (<5 h post invasion) were incubated at 5% parasitemia and 2% hemato-crit. Trophozoites were harvested at 20 h post incubation. Red blood cells were lysed with 0.05% saponin followed by multiple washes with PBS to remove traces of hemoglobin. The pellet was then resuspended in PBS and cell number quantified by flow cytometry. The remaining parasites were then lysed using hypotonic stress and sonication. Following centrifugation at 3600 r.p.m. for 20 min at RT, treatment with 0.2 M HEPES buffer (pH 7.4), 4% SDS (w/v), 0.3 M NaCl, 25% pyridine, 0.3 M NaOH, 0.3 M HCl, and distilled water, the fractions corresponding to hemoglobin, free heme, and hemozoin were recovered as solution. The UV–visible spectrum of each heme fraction as an Fe(III)heme–pyridine complex was measured using a multi-well plate reader (Spectramax 340PC; Molecular Devices). The total amount of each heme species was quantified using a standard curve prepared from a standard solution of porcine hematin (Sigma Aldrich) serially diluted in the same solvents used to process the pellets. The mass of each heme species in femtogram per cell (fg/cell) was then calculated by dividing the total amount of each heme species by the corresponding number of parasites. Statistical comparisons were made using one-way ANOVA (GraphPad Software Inc., La Jolla, CA, USA).

**Statistics and reproducibility**. All biological replicates were performed with independently derived parasite populations. Continuous variable data were presented as the mean ± SD unless indicated. Where indicated, statistical tests were performed on raw data, prior to normalization. The significance of differences was assessed by unpaired, two-tailed Student's $t$-test. A one- or two-way ANOVA followed by Sidak's or Dunnett's multiple comparison tests were used where appropriate when the mean values of more than three groups were compared. $P$-values of <0.05 were defined as significant. All statistical analyses were performed and visualized by GraphPad Prism 8 (GraphPad Software Inc., La Jolla, CA, USA). Tables and results were visualized in Excel (Microsoft). All representative experiments were performed at least twice with comparable results.

**Reporting summary**. Further information on research design is available in the Nature Research Reporting Summary linked to this article.

## Data availability
CRISPR screening raw data are available through the Gene Expression Omnibus with the accession number GSE153785. Minimally processed CRISPR screen data are available in Supplementary Data 1. Raw thermal proteome profiling data are available through the ProteomeXchange Consortium via the PRIDE partner repository[127] with the dataset identifier PXD019917. Minimally processed thermal proteome profiling data is available in Supplementary Data 5. Additional data is available from the authors upon request. *T. gondii* genome information can be found in ToxoDB. Source data are provided with this paper.

## Code availability
All software used in this study is commercially available or sourced from cited publications.

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

## Acknowledgements
We would like to thank the MIT Genome Core and the Whitehead Metabolomics Core for technical assistance. Emily Shortt and Eric Spooner assisted with proteomics. Sheena Vasquez and Jade Bath helped with assay development. George Bell provided statistics advice. This study was supported by Sir Henry Wellcome and Sir Henry Dale fellowships (103972/Z/14/Z, 213455/Z/18/Z) to C.R.H., a Robert Black Fellowship from the Damon Runyon Cancer Research Foundation (DRG-2365-19) to E.A.B., a National Science Foundation Graduate Research Fellowship (174530) to A.L.H., an NIH R01 to D.A.F. (AI109023), a Discovery Award from the US Department of Defense (W81XWH1910086) to D.A.F., and an NIH Director's Early Independence Award (1DP5OD017892) and a grant from the Mallinckrodt Foundation to S.L.

## Author contributions
Conceptualization, C.R.H., S.M.S., B.P., and S.L. Resources, B.M.M., A.L.H., and E.A.B. Investigation, C.R.H., S.M.S., B.P., A.L.H., N.F.G., J.O., and K.W. Writing–original draft, C.R.H., S.M.S., and B.P. Writing–review and editing, C.R.H., S.M.S., B.P., N.F.G., D.A.F., and S.L. Funding acquisition, D.A.F. and S.L. Supervision, D.A.F. and S.L.

## Competing interests
The authors declare no competing interests.
