## [Peer Review File · Nature Communications]

Reviewers' Comments:

Reviewer #1:

Remarks to the Author:

Harding et al deploy a gene-editing screen approach in the tractable Apicomplexan *Toxoplasma gondii* to explore both K13-dependent and independent mechanisms of reduced artemisinin susceptibility. This is a sensible and useful approach and throws a spotlight on mechanisms centred on key mitochondrial functions.

The work is original and of general interest to the field, with clear relevance to studies of artemisinin susceptibility in *Plasmodium* spp.

My only major comment is that the authors have not more carefully explained the important differences between *T. gondii* and *P. falciparum*, and between different variants of the K13-encoding genes in the two species investigated in the paper. Thus some potential weaknesses in the data that may confound the conclusions drawn are not explicitly stated.

In this Reviewer's opinion this can be overcome by a clear paragraph in the Discussion setting out the following caveats

- artemisinin is not used as a treatment option for toxoplasmosis as it is for malaria. Is this because the level of dependence on haemoglobin metabolism in *Plasmodium* spp. (being intra-erythrocytic) is much higher than for *Toxoplasma* spp. (being promiscuous in host cell requirement)? I thought so. Then this represents major differences in cell biology that mean comparative mutagenesis studies need to be interpreted with caution.

- I am uncomfortable with comparison of Tg edited at the Pf codon 580 orthologous position to Pf edited at the codon 539 position of the K13 gene. Are Cam 3.11 engineered with the 580Y allele not available, as in the studies of Stramer? This difference needs to be acknowledged or extra experiments done with the C580Y equivalent. This is also relevant to the Figure 6 experiments in panels d to h. Why was the Cam3.11 C580Y variant not included here? It is known to have a less extreme phenotype to that of the R539T mutation.

Minor comments:

- page 9, second paragraph. This Reviewer's understanding of the studies of Klonis and colleagues (ref 38) is that (in wild-type *P. falciparum*) there is a brief window of high artemisinin susceptibility in the first few hours post-invasion, which then falls as the ring-stage trophozoite matures, then rises during schizogony (see his Figure 1). Is the Hb available to activate artemisinin at this stage certain to be mitochondrial only? Is not the apicoplast also a possible source?

- throughout, nomenclature is not fully compliant with antimicrobial chemotherapy convention: "susceptibility" of a pathogen to drug should be used instead of "sensitivity"; EC50 (effective conc) is preferable to IC50 as it encompasses both inhibitory and cytotoxic effects, which cannot be readily distinguished in most assays deployed.

Reviewer #2:

Remarks to the Author:

This study explores genetic determinants of dihydroartemisinin (DHA) sensitivity by *Toxoplasma gondii* parasites. The authors use rational mutagenesis and CRISPR-based screens to identify genes whose mutation or disruption either increases or decreases DHA sensitivity by *T. gondii*. Based on multiple gene connections to heme synthesis, they show that chemical inhibitors of heme synthesis or the TCA

cycle also confer DHA resistance. Finally, the authors present data that a chemical inhibitor of heme synthesis or mitochondrial DegP gene deletion in *Plasmodium falciparum* also modulate DHA

sensitivity in malaria parasites. The authors conclude that distinct Apicomplexan parasites can have common mechanisms of DHA resistance.

Artemisinins are current frontline antimalarial treatments also under development for treating other infectious diseases and cancer. Thus, understanding mechanisms of artemisinin activation and resistance is of substantial importance, with large potential impact on treatment and resistance prevention. The main significance of this manuscript is identification of novel genetic loci in *T. gondii* that affect DHA sensitivity. Connections identified in the paper between mitochondrial heme synthesis and DHA activation in *T. gondii* are consistent with general expectations that DHA requires activation by heme and that mitochondrial synthesis is the dominant source of heme in *T. gondii*. However, these mechanistic connections are not developed in depth, and key doubts remain about direct versus indirect effects that weaken overall conclusions. The *P. falciparum* studies are weak and unconvincing.

1. DHA has 10-100 fold weaker activity against *T. gondii* (IC₅₀ 70-550 nM) compared to *P. falciparum* (IC₅₀ ~7 nM). DHA is also thought to be predominantly activated in Plasmodium by heme released from host hemoglobin digestion, which is not a feature of Toxoplasma biology. In the introduction, it seems misleading to motivate study of DHA activity in *T. gondii* as potentially revealing pan-Apicomplexan mechanisms of DHA resistance and activation without mentioning these differences.

2. Throughout the text, the authors make qualitative comparisons in DHA sensitivity and porphyrin/heme levels (e.g., "lowered levels of free heme and decreased DHA susceptibility"), even though the figures and tables supply quantitative IC₅₀ values and relative metabolite levels. Qualitative comparisons make it difficult to gauge the magnitude of effects, and textual comparisons would be substantially strengthened and clarified by making quantitative comparisons where possible (e.g., the IC₅₀ value increased X-fold from value Y to value Z).

3. Analysis of relative metabolite levels in untreated parasites in Figures 3, 4, S2, and S3 is misleading. Parental/untreated values are normalized to 100% without error bars, giving the appearance that there is no uncertainty/variation in metabolite values for these samples. It is unclear what comparisons were made to determine statistical significance. Was 2-way ANOVA performed prior to or after normalization of parental/untreated values? For transparency, it would seem preferable to express all samples either as the absolute amount/cell (as in Fig. 6) or as the relative intensity compared to internal standard (including average and SD for parental/untreated samples).

4. TMEM14C was suggested in ref. 42 in mammalian cells to import protoporphyrinogen IX into the mitochondrial matrix, based on accumulation of upstream porphyrins and diminished PPIX and heme in a TMEM14C KO. If the *T. gondii* homolog has a similar function, one would expect a similar reduction in heme synthesis in the TGGT1_228110 KO, which would be expected to reduce DHA activation and thus decrease DHA sensitivity based on the authors' model. However, the KO increases DHA sensitivity in *T. gondii* and does not cause significant changes in parasite heme or PPIX levels (Fig. S2). These contradictions raise substantial doubts regarding the function of this gene in *T. gondii*. Thus, the mechanism by which the Δ TMEM14C KO affects DHA sensitivity in *T. gondii* does not seem at all clear.

5. Does disruption of PBGD, PPOX, and/or TCA enzymes reduce heme synthesis and DHA sensitivity? Increased drug scores for these mutants suggest DHA resistance, which the authors interpret as due to decreased heme synthesis, but no data in the paper clearly establish that either change is observed. These genes may be essential, which would complicate testing stable KO's, but a conditional (e.g., Tet system) knock-down of one of these proteins (e.g., PPOX) and demonstration of DHA resistance would substantially strengthen the conclusion that heme synthesis modulates DHA sensitivity.

6. 10 mM succinylacetone used by the authors in *T. gondii* seems enormously high and brings into doubt if the change in DHA sensitivity is due to on- or off-target effects. In *Plasmodium*, SA has off-target activity/toxicity at concentrations >500 μ M (Nagaraj, PLoS Pathog., 2013). This concern makes a conditional knock-down of PPOX (or PBGD) more critical.

7. If the growth defects of Δ DegP2 in *T. gondii* cannot be complemented by a WT DegP2 copy, what is the basis for concluding that the reduction in DHA sensitivity in Δ DegP2 is due to the observed reduction in heme in that mutant rather than some confounding off-target genetic change? Does the Δ DegP2 + DegP2-HA line have restored heme levels equivalent to WT? The authors assess total porphyrins in Fig. 4e but heme is the critical analyte and the complement line should be tested in Fig. 4g to more directly link DegP2 function to heme levels.

8. What is the function and/or substrate(s) of DegP2 in Apicomplexa, and what is the mechanism by which DegP2 might impact heme levels? The authors offer no hypothesis on this point. Up-regulated expression of heme-binding ETC subunits encoded by the mitochondrial genome in Δ DegP2 parasites would most simply suggest enhanced heme synthesis to furnish the cofactor for these subunits. The authors, however, report diminished heme levels for the Δ DegP2 mutant, which is confusing. The authors offer no explanation to reconcile these contrasting observations.

Nine. Succinylacetone has well documented off-target toxicity in *P. falciparum* (Nagaraj, PLoS Pathog., 2013 and Ke, JBC, 2014), raising doubts if the small \sim 2-fold change in DHA sensitivity in *Plasmodium* due to 200 μ M SA is from diminished heme synthesis or off-target effects. Heme synthesis is not essential in blood-stage *P. falciparum*, and multiple enzyme KO's (e.g., ALAS, FECH, CPOX, etc.) are available in the community (e.g., Ke, JBC, 2014). Do these KO parasites show DHA tolerance by RSA? No change in DHA IC50 was observed for the ALAS and FECH KO's in *P. falciparum* (Ke, JBC, 2014).

10. Transcription of heme synthesis enzymes in *P. falciparum* only commences after 15-20 hours post-invasion (Stunnenberg 3D7 RNA-Seq data from PlasmoDB) in trophozoites, suggesting that heme synthesis is not active in rings and raising doubts that the small impact of SA on DHA sensitivity by RSA is due to diminished heme synthesis.

11. Related to #10, what is the basis for the authors' statement in the Discussion (bottom, page nine) that "*P. falciparum* rings ... appear to derive their heme mainly from mitochondrial pathways"? The cited reference 80 makes the opposite conclusion that hemoglobin digestion and heme release begins in rings, a conclusion further supported by later publications, including Heller and Roepe, Biochem., 2018 and Tilley et al., J Cell Sci, 2016. This later reference reported reduced DHA sensitivity by RSA in mutants of falcipain 2, a food vacuole protease, as expected if hemoglobin-derived heme is the dominant activator of DHA in rings. The authors also cite ref. 4, but this study has multiple flaws, including use of 500 μ M SA (a concentration with documented off-target toxicity-see above) and studies of ALA effects on DHA labeling in ring-stage parasites, even though rings do not take up ALA as they lack the NPP pathways upon which ALA uptake depends (ref. 53), suggesting off-target effects.

12. Differences in heme levels reported for WT and mutant *P. falciparum* parasites in Fig. 6F may be statistically significant (based on 2 measurements), but these differences are not substantial and do not support a strong conclusion that differences in DHA sensitivity by RSA derive uniquely from variable heme levels.

13. Why is there such a large variation (>10-fold) in 0-3h DHA RSA in WT parasites in 6A vs 6D? This large >10-fold variation contradicts the tight <2-fold variations reported in each individual assay. This

large inconsistency is worrisome, especially since RSA survival of Δ DegP parasites in 6D is within 2-fold of what should be identical measurement of WT sensitivity in 6A, raising doubts about the effect of DegP KO on DHA tolerance in *P. falciparum*.

Reviewer #3:

Remarks to the Author:

Harding et al., Tg ART screen

This is an elegant study seeking to uncover genetic factors potentially related to *Toxoplasma* and *Plasmodium* sensitivity to artemisinin, especially factors that may underlie emerging artemisinin resistance in *Plasmodium falciparum*. The authors employ a number of sophisticated tools, which includes whole genome screens of *Toxoplasma* CRISPR mutants exposed to sublethal and lethal concentrations of DHA. These phenotype screens confirmed known associations and identified new factors in the parasites' biosynthetic pathways that relative heme abundance regulates DHA sensitivity in apicomplexans. These primary findings reinforce a consensus in the field that free heme has a central role in regulating ART sensitivity. The novel discovery of the study is the identification of mitochondrial metabolic processes potentially important in regulating heme abundance and by inference sensitivity to artemisinin killing. Importantly, this implicates heme not derived by hemoglobin digestion as important in activating artemisinin parasitocidal activity.

The newly identified processes include a putative inner membrane heme transporter, *Tmem14c*, and a putative serine protease involved in processing mitochondrial membrane proteins associated with TCA/electron transport complex. These are significant discoveries in understanding heme metabolism in apicomplexans and potentially important for providing insights into design of new artemisinin combination therapies. Therefore, the study adds a potentially important new dimension to understanding artemisinin mechanisms of action for killing apicomplexans and how they develop resistance to this parasitocidal activity.

Equally important is the methodological advance in utilizing a whole genome forward genetic screen for an apicomplexan species to identify genetic factors associated with a selected phenotype. This approach coupled with more traditional targeted mutagenesis and pharmacological approaches represents a powerful new methodology to experimentally query the *Toxoplasma* genome. Overall, it is an impressive with important knowledge of basic biological and clinical significance gained.

Major concerns:

1. The failure to wholly complement DegP2 mutant created from the CRISPR screen indicates there are unaccounted for additional defects that occurred during the mutagenesis. While generation and functional characterization of a DegP2 KO indicates this is likely the main principal genetic mutation for the observed phenotype, the incomplete characterization of the defect(s) undermines confidence in the direct phenotype-genotype association and conclusions drawn from functional characterization of the mutant. This would not be so important if the functional characterization did not have prominence in the main conclusions of the study.
2. The high dose screen reported identifying 73 genes important in regulating DHA sensitivity and TCA enzymes were enriched in those identified. Of these 73 genes 65 were not confirmed in independent biological replicates and should not be included without some type of additional independent validation of the genotype-phenotype link. Also, it is implied but poorly justified in the background and experimental results provided that the function of the *P. falciparum* TCA cycle is equivalent to that of *Toxoplasma* – this conclusion should be supported better, connecting the dots is needed.
3. What is the relevance of the study's findings to artemisinin resistance in field isolates of *P. falciparum*? The study demonstrates that an apicomplexan's intracellular heme concentration is linked with its sensitivity to artemisinin and a main implication is heme biosynthesis, especially from the

mitochondrion has an important clinical significance in malaria. Therefore, an important implication of these studies is heme outside of the food vacuole (i.e., in the parasite cytoplasm and possibly elsewhere) plays a critical role in activation of artemisinin and regulating its parasitocidal activity. However, this conclusion seems to be undermined by the last set of experiments, demonstrating artemisinin sensitivity of the PfDegP mutant and Cam3.II during ring-stage development. Generally, ring-stages are considered clinically resistant to artemisinin. Perhaps, I have only misinterpreted the authors' message and this section simply should be revised with a clearer message.

Minor comments:

1. Did the authors analyze other kelch genes for mutations?
2. The apparent K13 phenotypes of the Pf and Tg K13 mutants is interesting and suggestive of similar functions in these very different parasites. However, given the currently poor understanding of exactly what does K13 do in *P. falciparum* and also how K13 mutations confer resistance/delayed clearance, the conclusions of functional equivalence remain overly speculative.
3. WGS of the DegP2 might answer what other genetic changes occurred in generating this mutant and provide additional understanding of genetic factors that can be linked to altered DHA sensitivity phenotypes.
4. What is the % coverage of the genome by the CRISPR mutagenesis method used.
5. The manuscript uses a lot of technical jargon that lacks clear meaning for those not in this field of study (for example, "guide RNAs... were enriched").

RESPONSE TO REVIEWERS

*Author responses highlighted in blue

Reviewer #1 (Remarks to the Author):

Harding et al deploy a gene-editing screen approach in the tractable Apicomplexan *Toxoplasma gondii* to explore both K13-dependent and independent mechanisms of reduced artemisinin susceptibility. This is a sensible and useful approach and throws a spotlight on mechanisms centred on key mitochondrial functions.

The work is original and of general interest to the field, with clear relevance to studies of artemisinin susceptibility in *Plasmodium* spp.

My only **major comment** is that the authors have not more carefully explained the important differences between *T. gondii* and *P. falciparum*, and between different variants of the K13-encoding genes in the two species investigated in the paper. Thus some potential weaknesses in the data that may confound the conclusions drawn are not explicitly stated.

In this Reviewer's opinion this can be overcome by a clear paragraph in the Discussion setting out the following caveats

- artemisinin is not used as a treatment option for toxoplasmosis as it is for malaria. Is this because the level of dependence on haemoglobin metabolism in *Plasmodium* spp. (being intra-erythrocytic) is much higher than for *Toxoplasma* spp. (being promiscuous in host cell requirement)? I thought so. Then this represents major differences in cell biology that mean comparative mutagenesis studies need to be interpreted with caution.

We appreciate the reviewer's positive evaluation of our work. It was not our intention to overrepresent the similarities between *Toxoplasma* and *Plasmodium*, but thought it was important to point out the unexpected similarities that emerge from our work through the analysis of Kelch 13 mutations in *Toxoplasma* and DegP2 (PfDegP) loss in *Plasmodium*. We have attempted to capture these differences by including the following statements:

"we recognize that *T. gondii* is far less sensitive to DHA than blood-stage malaria parasites, a fact that contributes to the use of other compounds as front-line drugs for toxoplasmosis"

"These observations help explain why blood-stage *P. falciparum*, releasing large amounts of heme from the digestion of hemoglobin, is more susceptible to artemisinin than *T. gondii*⁷⁵⁻⁷⁷. Interestingly, *Babesia* spp., which live within erythrocytes but do not take up hemoglobin, have an intermediate sensitivity to artemisinin^{78,79}, while *Cryptosporidium parvum*—which lacks genes necessary for heme biosynthesis^{80,81}—shows little response to artemisinin⁸²."

We also discuss differences in the balance between heme scavenging and biosynthesis between the two species, after which we state "Our results indicate that there are important parallels between *T. gondii* and *P. falciparum* responses to DHA, despite *T. gondii*'s reduced susceptibility to such compounds."

We finally conclude stating, "Despite critical metabolic differences and over 350 million years of divergent evolution¹¹¹, our screens identified multiple genes involved in heme biosynthesis as critical determinants

of DHA susceptibility in *T. gondii*, echoing the results of recent studies that demonstrate that hemoglobin import greatly affects *P. falciparum*'s response to artemisinin^{14,15}."

- I am uncomfortable with comparison of Tg edited at the Pf codon 580 orthologous position to Pf edited at the codon 539 position of the K13 gene. Are Cam 3.11 engineered with the 580Y allele not available, as in the studies of Straimer? This difference needs to be acknowledged or extra experiments done with the C580Y equivalent. This is also relevant to the Figure 6 experiments in panels d to h. Why was the Cam3.11 C580Y variant not included here? It is known to have a less extreme phenotype to that of the R539T mutation.

As pointed out by the reviewer, differences between the C580Y and R539T mutations have already been explored in the literature (Straimer et al. 2015. *Science*). In our experiments, the R539T is therefore used as a positive control for a mutant that shows decreased sensitivity to DHA. The relevant comparison is between the parental strain and the *PfDegP* knockout, which supports the conclusion that loss of *PfDegP* modestly, but significantly, reduces sensitivity to DHA.

Minor comments:

- page 9, second paragraph. This Reviewer's understanding of the studies of Klonis and colleagues (ref 38) is that (in wild-type *P. falciparum*) there is a brief window of high artemisinin susceptibility in the first few hours post-invasion, which then falls as the ring-stage trophozoite matures, then rises during schizogony (see his Figure 1). Is the Hb available to activate artemisinin at this stage certain to be mitochondrial only? Is not the apicoplast also a possible source?

As shown below in Fig. 1 from Klonis et al. (2013. *PNAS*) the pattern of DHA susceptibility is complex. More recent studies have pointed to the role of hemoglobin digestion in increased DHA susceptibility (Birnbaum et al. 2020. *Science*). It is likely that the availability of free heme is not the sole determinant of DHA susceptibility; the presence of targets for alkylation and pathways to repair or overcome damage from alkylation will also vary across erythrocytic stages and influence DHA susceptibility. DHA susceptibility is therefore an imperfect correlate of heme availability. While several studies have shown that biosynthetic pathways remain active in blood stages, they are clearly dispensable, making the precise contribution of *de novo* biosynthesis to total pools of heme unclear.

[Redacted]

Regarding the apicoplast, the current model for the heme biosynthesis pathway in *Toxoplasma* and *Plasmodium* places intermediates in the pathway within the apicoplast, but the final two enzymes (protoporphyrinogen oxidase and ferrochelatase) reside in the mitochondrion. Since it is the iron in heme that is thought to mediate the activation of DHA, we expect that the apicoplast is a source for intermediates in the pathway but not heme itself.

- throughout, nomenclature is not fully compliant with antimicrobial chemotherapy convention: "susceptibility" of a pathogen to drug should be used instead of "sensitivity"; EC50 (effective conc) is preferable to IC50 as it encompasses both inhibitory and cytotoxic effects, which cannot be readily distinguished in most assays deployed.

We appreciate the reviewers recommendations and have modified all relevant references to “sensitive/sensitivity” and “EC₅₀”, as suggested.

Reviewer #2 (Remarks to the Author):

This study explores genetic determinants of dihydroartemisinin (DHA) sensitivity by *Toxoplasma gondii* parasites. The authors use rational mutagenesis and CRISPR-based screens to identify genes whose mutation or disruption either increases or decreases DHA sensitivity by *T. gondii*. Based on multiple gene connections to heme synthesis, they show that chemical inhibitors of heme synthesis or the TCA cycle also confer DHA resistance. Finally, the authors present data that a chemical inhibitor of heme synthesis or mitochondrial DegP gene deletion in *Plasmodium falciparum* also modulate DHA sensitivity in malaria parasites. The authors conclude that distinct Apicomplexan parasites can have common mechanisms of DHA resistance.

Artemisinins are current frontline antimalarial treatments also under development for treating other infectious diseases and cancer. Thus, understanding the mechanisms of artemisinin activation and resistance is of substantial importance, with large potential impact on treatment and resistance prevention. The main significance of this manuscript is identification of novel genetic loci in *T. gondii* that affect DHA sensitivity. Connections identified in the paper between mitochondrial heme synthesis and DHA activation in *T. gondii* are consistent with general expectations that DHA requires activation by heme and that mitochondrial synthesis is the dominant source of heme in *T. gondii*. However, these mechanistic connections are not developed in depth, and key doubts remain about direct versus indirect effects that weaken overall conclusions. The *P. falciparum* studies are weak and unconvincing.

We thank the reviewer for their critical evaluation of our work, and have attempted to provide further mechanistic details about the connection between DegP2 and mitochondrial metabolism. However, the precise functions of Tmem14c and DegP2 would require far more work than we can reasonably include in this manuscript and have not been trivial to define. We believe that the correlation between the multiple pathways and heme availability is sufficiently strong in aggregate to conclude that the newly characterized loci likely modify DHA sensitivity in a similar manner. We should note that discovering new loci associated with DHA or artemisinin resistance has not traditionally been accompanied with a precise understanding of the mechanisms involved; mutations in Kelch13 were known to cause mutations years before any mechanistic explanation was developed, and more recently mutations in Coronin were reported to decrease DHA susceptibility although the mechanism remains unknown. Therefore, we would request that similar standards be extended to our study.

1. DHA has 10-100 fold weaker activity against *T. gondii* (IC₅₀ 70-550 nM) compared to *P. falciparum* (IC₅₀ ~7 nM). DHA is also thought to be predominantly activated in *Plasmodium* by heme released from host hemoglobin digestion, which is not a feature of *Toxoplasma* biology. In the introduction, it seems misleading to motivate study of DHA activity in *T. gondii* as potentially revealing pan-Apicomplexan mechanisms of DHA resistance and activation without mentioning these differences.

It was not our intention to mislead the reader and, as noted in response to Reviewer 1, we have now more carefully expressed the differences between the two species, stating that both DHA susceptibility and the balance between heme salvage and de novo biosynthesis present significant differences between *Toxoplasma* and *Plasmodium*. It is worth mentioning that differences in permeability, compensatory pathways, stress responses, and even the precise affinity of a drug target, can all influence the EC₅₀ of a

compound in question, such that differences in susceptibility do not formally exclude the presence of conserved mechanisms of drug resistance or activation. Nevertheless, we have included the following statement in the Results section, where we discuss the susceptibility of *T. gondii* to DHA:

“we recognize that *T. gondii* is far less sensitive to DHA than blood-stage malaria parasites, a fact that contributes to the use of other compounds as front-line drugs for toxoplasmosis”

2. Throughout the text, the authors make qualitative comparisons in DHA sensitivity and porphyrin/heme levels (e.g., “lowered levels of free heme and decreased DHA susceptibility”), even though the figures and tables supply quantitative IC50 values and relative metabolite levels. Qualitative comparisons make it difficult to gauge the magnitude of effects, and textual comparisons would be substantially strengthened and clarified by making quantitative comparisons where possible (e.g., the IC50 value increased X-fold from value Y to value Z).

We have modified the text to provide references to fold changes, and precise EC₅₀ values. All DHA EC₅₀ values are also provided in Supplementary Table 1.

3. Analysis of relative metabolite levels in untreated parasites in Figures 3, 4, S2, and S3 is misleading. Parental/untreated values are normalized to 100% without error bars, giving the appearance that there is no uncertainty/variation in metabolite values for these samples. It is unclear what comparisons were made to determine statistical significance. Was 2-way ANOVA performed prior to or after normalization of parental/untreated values? For transparency, it would seem preferable to express all samples either as the absolute amount/cell (as in Fig. 6) or as the relative intensity compared to internal standard (including average and SD for parental/untreated samples).

Due to high variability in heme measurements obtained from mass spectrometry, we have omitted these results from the revised manuscript. Porphyrin measurements for the new Figures 5 and 7, and Supplementary Figures 2 and 3 are now expressed in absolute terms from a fixed number of cells as described in the materials and methods.

4. TMEM14C was suggested in ref. 42 in mammalian cells to import protoporphyrinogen IX into the mitochondrial matrix, based on accumulation of upstream porphyrins and diminished PPIX and heme in a TMEM14C KO. If the *T. gondii* homolog has a similar function, one would expect a similar reduction in heme synthesis in the TGGT1_228110 KO, which would be expected to reduce DHA activation and thus decrease DHA sensitivity based on the authors’ model. However, the KO increases DHA sensitivity in *T. gondii* and does not cause significant changes in parasite heme or PPIX levels (Fig. S2). These contradictions raise substantial doubts regarding the function of this gene in *T. gondii*. Thus, the mechanism by which the Δ TMEM14C KO affects DHA sensitivity in *T. gondii* does not seem at all clear.

The reviewer correctly summarizes the current model for TMEM14C function during hematopoiesis, as defined by Yien et al. However, it is important to note that the specific properties of the putative transporter have not been examined, and TMEM14c lacks motifs that might specify its directionality. We therefore now state in the Discussion that, “Although we could not establish a direct role for Tmem14c as a porphyrin transporter and cannot formally exclude alternative roles in mitochondrial metabolism, several lines of evidence lead us to propose that Tmem14c transports heme out of the mitochondrion in *T. gondii*.” We also note that “the mechanism and substrate specificity of TMEM14C remain uncharacterized in mammalian cells, leaving open the possibility that TMEM14C might simply mediate passive transport of porphyrins down their concentration gradient.”

Based on the similarities between *T. gondii* and *Plasmodium* spp. heme biosynthesis pathways, and the absence of Tmem14c from *Plasmodium* spp., Tmem14c is unlikely to be the major means of transporting PPIX into the mitochondrion since we would expect such a function to be conserved between the two genera. As pointed out by the reviewer, there is no significant change in total porphyrin levels resulting from the loss of Tmem14c, such that the biosynthetic pathway doesn't seem perturbed, but we cannot rule out changes in the distribution of heme throughout the cell. The accumulation of heme in the mitochondrion therefore remains the most parsimonious explanation for the increased DHA sensitivity, but we agree with the reviewer that further study will be necessary to demonstrate this mechanistically.

5. Does disruption of PBGD, PPOX, and/or TCA enzymes reduce heme synthesis and DHA sensitivity? Increased drug scores for these mutants suggest DHA resistance, which the authors interpret as due to decreased heme synthesis, but no data in the paper clearly establish that either change is observed. These genes may be essential, which would complicate testing stable KO's, but a conditional (e.g., Tet system) knock-down of one of these proteins (e.g., PPOX) and demonstration of DHA resistance would substantially strengthen the conclusion that heme synthesis modulates DHA sensitivity.

While we do not directly disrupt the genes involved in the TCA cycle or heme biosynthesis, we do employ inhibitors of these pathways (Figure 3), which significantly reduce total porphyrin concentrations and DHA susceptibility. We do not directly knock out these enzymes, because as the reviewer points out, *T. gondii* deficient in heme biosynthesis or the TCA cycle show substantially reduced fitness. However, Fig. 5e correlates porphyrin concentrations and DHA susceptibility across several different mutants, further strengthening the relationship between these two phenotypes.

Following the reviewer's recommendation, we obtained a Tet-inducible knockdown of Ferrochelatase (Bergman et al. 2020. *PLoS Pathogens*); however, by the time knockdown was achieved the substantial loss in parasite viability made it impossible for us to determine an EC₅₀ for DHA. Because these enzymes are not accessible to tunable post-transcriptional regulation systems, chemical inhibition, as described above, remains the best approach to establish their function.

6. 10 mM succinylacetone used by the authors in *T. gondii* seems enormously high and brings into doubt if the change in DHA sensitivity is due to on- or off-target effects. In *Plasmodium*, SA has off-target activity/toxicity at concentrations >500 μ M (Nagaraj, *PLoS Pathog.*, 2013). This concern makes a conditional knock-down of PPOX (or PBGD) more critical.

Due to the reviewer's concerns about off-target effects we have removed the experiments using this inhibitor in *Plasmodium*.

Although we use a higher dose of SA (10 mM) for the *T. gondii* experiments than that associated with *Plasmodium* off-target effects, we do not see diminished growth—presumably because sufficient heme is still produced or scavenged, or because downstream heme intermediates can be scavenged, as suggested for *P. falciparum* (Sigala et al. 2015. *Elife*) and *T. gondii* (Krishnan et al. 2020. *Cell Host Microbe*). SA treatment led to the expected decrease in total porphyrin concentrations (Fig. 3f) and modest changes in the polar metabolites betaine and ornithine (Supplementary Fig. 3a), which are consistent with subtle changes in mitochondrial metabolism. There is therefore no evidence that SA has off-target effects in *T. gondii* at the concentrations used. Moreover, inhibition of the speculative off-target would have to protect parasites against DHA, which seems improbable, whereas the on-target effects of the compound is consistent with the extensive additional data that implicates heme concentrations in this process.

7. If the growth defects of Δ DegP2 in *T. gondii* cannot be complemented by a WT DegP2 copy, what is the basis for concluding that the reduction in DHA sensitivity in Δ DegP2 is due to the observed reduction in heme in that mutant rather than some confounding off-target genetic change? Does the Δ DegP2 + DegP2-HA line have restored heme levels equivalent to WT? The authors assess total porphyrins in Fig. 4e but heme is the critical analyte and the complement line should be tested in Fig. 4g to more directly link DegP2 function to heme levels.

We agree with the reviewer's concern and have addressed the issue by constructing a new conditional mutant of DegP2 (Figs. 4–6). Consistent with the original Δ DegP2 harboring unrelated changes that contributed to reduced fitness, neither the catalytic mutant (DegP2^{S569A}-Ty) nor the conditional knockdown (cKD +Rapa) showed reduced plaque formation (Figs. 5b and 5f). Conditional knockdown of DegP2 recapitulated the reduction in total porphyrin concentrations observed in the knockout along with the reduction in DHA susceptibility (Figs. 5g–h).

Because knockdown of DegP2 had no perceptible effect on parasite fitness, we were able to directly compare its effect during DHA treatment using competition assays (Fig. 5h) which ensures a more direct comparison to the wild-type strain. These results allow us to conclude that the effects on porphyrin levels and DHA susceptibility were indeed attributable to loss of DegP2 in the Δ DegP2.

8. What is the function and/or substrate(s) of DegP2 in Apicomplexa, and what is the mechanism by which DegP2 might impact heme levels? The authors offer no hypothesis on this point. Up-regulated expression of heme-binding ETC subunits encoded by the mitochondrial genome in Δ DegP2 parasites would most simply suggest enhanced heme synthesis to furnish the cofactor for these subunits. The authors, however, report diminished heme levels for the Δ DegP2 mutant, which is confusing. The authors offer no explanation to reconcile these contrasting observations.

We agree with the reviewer that identifying the substrate(s) of DegP2 is a fascinating research direction. We present new data using thermal proteome profiling to identify proteins that change in their thermal stability when DegP2 is depleted. Using this approach, we identified three mitochondrial proteins—the NifU domain-containing protein TGGT1_212930, the ATP synthase γ subunit, and the un-annotated protein TGGT1_226500. We are particularly interested in TGGT1_212930 because in other systems, NifU domain-containing proteins transfer iron-sulfur clusters to Complex II in the electron transport chain, as well as to the TCA cycle enzyme aconitase (Melber et al. 2016. *Elife*). We show that Δ DegP2 parasites are less sensitive to the Complex II inhibitor TTFA than parental or Δ DegP2/DegP2-HA parasites. We consider this to be strong evidence that DegP2 interacts with Complex II, perhaps by chaperoning the iron-sulfur cluster transfer from TGGT1_212930 to SDHB. Loss of DegP2 would therefore impair the TCA cycle and the ETC thereby lowering heme biosynthesis, as we demonstrated through chemical inhibition of the TCA cycle. Analysis of polar metabolites showed changes in TCA intermediates consistent with perturbing the TCA cycle. The interconnectedness of the heme availability, the TCA cycle, and the ETC prevents us from definitively stating the directionality of the effects, but the data we present will act as a foundation for the extensive work needed to define the molecular function of DegP2.

9. Succinylacetone has well documented off-target toxicity in *P. falciparum* (Nagaraj, PLoS Pathog., 2013 and Ke, JBC, 2014), raising doubts if the small ~2-fold change in DHA sensitivity in Plasmodium due to 200 μ M SA is from diminished heme synthesis or off-target effects. Heme synthesis is not essential in blood-stage *P. falciparum*, and multiple enzyme KO's (e.g., ALAS, FECH, CPOX, etc.) are available in the community (e.g., Ke, JBC, 2014). Do these KO parasites show DHA tolerance by RSA? No change in DHA IC50 was observed for the ALAS and FECH KO's in *P. falciparum* (Ke, JBC, 2014).

We have removed the *P. falciparum* data that relied on SA and limited our analysis to PfDegP.

10. Transcription of heme synthesis enzymes in *P. falciparum* only commences after 15-20 hours post-invasion (Stunnenberg 3D7 RNA-Seq data from PlasmoDB) in trophozoites, suggesting that heme synthesis is not active in rings and raising doubts that the small impact of SA on DHA sensitivity by RSA is due to diminished heme synthesis.

We have removed these data.

11. Related to #10, what is the basis for the authors' statement in the Discussion (bottom, page nine) that "P. falciparum rings ... appear to derive their heme mainly from mitochondrial pathways"? The cited reference 80 makes the opposite conclusion that hemoglobin digestion and heme release begins in rings, a conclusion further supported by later publications, including Heller and Roepe, *Biochem.*, 2018 and Tilley et al., *J Cell Sci*, 2016. This later reference reported reduced DHA sensitivity by RSA in mutants of falcipain 2, a food vacuole protease, as expected if hemoglobin-derived heme is the dominant activator of DHA in rings. The authors also cite ref. 4, but this study has multiple flaws, including use of 500 μ M SA (a concentration with documented off-target toxicity- see above) and studies of ALA effects on DHA labeling in ring-stage parasites, even though rings do not take up ALA as they lack the NPP pathways upon which ALA uptake depends (ref. 53), suggesting off-target effects.

We have removed this statement.

12. Differences in heme levels reported for WT and mutant *P. falciparum* parasites in Fig. 6F may be statistically significant (based on 2 measurements), but these differences are not substantial and do not support a strong conclusion that differences in DHA sensitivity by RSA drive uniquely from variable heme levels.

We have provided additional measurements to further show that free heme is lower in Δ PfDegP parasites than in wild-type parasites. We disagree with the reviewer's conclusion that a moderate difference in heme levels cannot alter DHA sensitivity. The relationship between heme and DHA sensitivity has not been studied in enough detail to know how much of a reduction in heme levels is necessary to cause a change in DHA susceptibility. In addition, our data reflect a change in bulk heme levels, which may be more pronounced in certain cellular compartments. Lastly, our *P. falciparum* data must be considered together with our evidence that lowering *T. gondii*'s heme levels, chemically or genetically, alters DHA sensitivity, and with evidence provided by others (Yang et al. 2019. *Cell Rep*; Birnbaum et al. 2020. *Science*) that draws similar conclusions. Considering all of these lines of evidence, we believe there is strong reason to think that variation in heme levels affects DHA sensitivity.

13. Why is there such a large variation (>10-fold) in 0-3h DHA RSA in WT parasites in 6A vs 6D? This large >10-fold variation contradicts the tight <2-fold variations reported in each individual assay. This large inconsistency is worrisome, especially since RSA survival of Δ DegP parasites in 6D is within 2-fold of what should be identical measurement of WT sensitivity in 6A, raising doubts about the effect of DegP KO on DHA tolerance in *P. falciparum*.

This discrepancy was the result of a clerical error, which we have now corrected. Thank you for pointing it out.

Reviewer #3 (Remarks to the Author):

Harding et al., Tg ART screen

This is an elegant study seeking to uncover genetic factors potentially related to *Toxoplasma* and *Plasmodium* sensitivity to artemisinin, especially factors that may underlie emerging artemisinin resistance in *Plasmodium falciparum*. The authors employ a number of sophisticated tools, which includes whole genome screens of *Toxoplasma* CRISPR mutants exposed to sublethal and lethal concentrations of DHA. These phenotype screens confirmed known associations and identified new factors in the parasites' biosynthetic pathways that relative heme abundance regulates DHA sensitivity in apicomplexans. These primary findings reinforce a consensus in the field that free heme has a central role in regulating ART sensitivity. The novel discovery of the study is the identification of mitochondrial metabolic processes potentially important in regulating heme abundance and by inference sensitivity to artemisinin killing. Importantly, this implicates heme not derived by hemoglobin digestion as important in activating artemisinin parasitocidal activity.

The newly identified processes include a putative inner membrane heme transporter, *Tmem14c*, and a putative serine protease involved in processing mitochondrial membrane proteins associated with TCA/electron transport complex. These are significant discoveries in understanding heme metabolism in apicomplexans and potentially important for providing insights into design of new artemisinin combination therapies. Therefore, the study adds a potentially important new dimension to understanding artemisinin mechanisms of action for killing apicomplexans and how they develop resistance to this parasitocidal activity.

Equally important is the methodological advance in utilizing a whole genome forward genetic screen for an apicomplexan species to identify genetic factors associated with a selected phenotype. This approach coupled with more traditional targeted mutagenesis and pharmacological approaches represents a powerful new methodology to experimentally query the *Toxoplasma* genome. Overall, it is an impressive work with important knowledge of basic biological and clinical significance gained.

Major concerns:

1. The failure to wholly complement *DegP2* mutant created from the CRISPR screen indicates there are unaccounted for additional defects that occurred during the mutagenesis. While generation and functional characterization of a *DegP2* KO indicates this is likely the main principal genetic mutation for the observed phenotype, the incomplete characterization of the defect(s) undermines confidence in the direct phenotype-genotype association and conclusions drawn from functional characterization of the mutant. This would not be so important if the functional characterization did not have prominence in the main conclusions of the study.

We agree with the reviewer, and so we have constructed an inducible *DegP2* mutant using the U1 system (*DegP2* cKD; Pieperhoff et al. 2015. *PLoS One*) and used this strain to confirm many of the results generated using our original Δ *DegP2* strain. *DegP2* cKD parasites form plaques normally after *DegP2* depletion, demonstrating that the growth defects originally reported for the Δ *DegP2* strain were indeed unrelated to loss of *DegP2*. Critically, porphyrin levels decrease in response to conditional depletion of *DegP2*, impacting DHA susceptibility, demonstrating that these phenotypes are indeed related to *DegP2* and supporting our main conclusions (Figs. 5f–h).

2. The high dose screen reported identifying 73 genes important in regulating DHA sensitivity and TCA enzymes were enriched in those identified. Of these 73 genes 65 were not confirmed in independent biological replicates and should not be included without some type of additional independent validation of the genotype-phenotype link. Also, it is implied but poorly justified in the background and experimental results provided that the function of the *P. falciparum* TCA cycle is equivalent to that of *Toxoplasma* – this conclusion should be supported better, connecting the dots is needed.

We have added the following paragraph to support our decision to use the full complement of 73 genes in our pathway analysis. We would also like to point out that this pathway analysis was only intended as a hypothesis generating tool, and that our more thorough analysis was based on genes that were reliably detected in multiple iterations of our screen.

“The likelihood of identifying a given candidate depends on the gene’s contribution to overall fitness as well as the gene’s impact on DHA susceptibility. For every iteration of the screen, the rate at which mutants are lost from the population will fluctuate such that certain fitness-conferring mutants may be completely lost from the population before they have a chance to impact survival under DHA treatment. Even candidates identified in a single screen are significant based on the concordant effect of multiple gRNAs; however, we have the highest confidence in hits obtained from multiple independent screens and focused subsequent analyses on these candidates.”

We have also added the following statement explaining how heme biosynthesis (which relies on the TCA cycle) differs between *T. gondii* and *P. falciparum*.

“*T. gondii* and *P. falciparum* differ in their reliance on *de novo* heme biosynthesis. Inhibiting heme biosynthesis either chemically⁸³ or genetically^{84,85} reduces the fitness of *T. gondii*, highlighting the importance of *de novo* heme biosynthesis to this parasite. By contrast, heme biosynthesis pathways are dispensable for *P. falciparum* growth during blood stages, although this pathway appears to be necessary during the mosquito stages^{86,87}. Although *de novo* heme synthesis is dispensable for blood stage *P. falciparum*, the components of this pathway are still expressed, and studies using radio-labelled substrates for heme biosynthesis have shown that the process remains active^{88–90}. Our results indicate that there are important parallels between *T. gondii* and *P. falciparum* responses to DHA, despite *T. gondii*’s reduced susceptibility to such compounds.”

3. What is the relevance of the study’s findings to artemisinin resistance in field isolates of *P. falciparum*? The study demonstrates that an apicomplexan’s intracellular heme concentration is linked with its sensitivity to artemisinin and a main implication is heme biosynthesis, especially from the mitochondrion has an important clinical significance in malaria. Therefore, an important implication of these studies is heme outside of the food vacuole (i.e., in the parasite cytoplasm and possibly elsewhere) plays a critical role in activation of artemisinin and regulating its parasitocidal activity. However, this conclusion seems to be undermined by the last set of experiments, demonstrating artemisinin sensitivity of the PfDegP mutant and Cam3.II during ring-stage development. Generally, ring-stages are considered clinically resistant to artemisinin. Perhaps, I have only misinterpreted the authors’ message and this section simply should be revised with a clearer message.

While ring stage *P. falciparum* are less sensitive to DHA than other stages of the lytic cycle, the Kelch 13 mutations associated with treatment failure alter DHA sensitivity precisely during the ring stage but have minimal impact thereafter (Ariey et al. 2014. *Nature*). DHA susceptibility is therefore commonly assessed by the ring-stage survival assay (Fig. 7c). A conclusion from our study is that mitochondrial sources of heme can be relevant to *P. falciparum* and *T. gondii* DHA susceptibility, which is supported by the data shown in Figure 7 and elsewhere. We have attempted to further clarify this message in the manuscript and

include a more thorough discussion of the difference between *T. gondii* and *P. falciparum*, which should provide more nuance to our conclusions.

Minor comments:

1. Did the authors analyze other kelch genes for mutations?

Currently, no other kelch genes have been associated with DHA susceptibility in *Plasmodium*, whereas large numbers of point mutations in K13 have been identified in clinical samples with altered susceptibility to DHA. In *T. gondii*, we chose to focus on one of the best characterized of these K13 mutations, C580Y. We have modified the text to make this more clear:

“In *P. falciparum*, point mutations in Kelch13 (K13), such as C580Y and R539T, correlate with delayed clearance and increased survival of ring-stage parasites^{12,13,38,39}. Although K13 is conserved among apicomplexans, its role in DHA susceptibility has not been examined in *T. gondii*. We chose to make a C627Y mutation in the *T. gondii* ortholog of K13 (TGGT1_262150), corresponding to *P. falciparum* C580Y”

2. The apparent K13 phenotypes of the Pf and Tg K13 mutants is interesting and suggestive of similar functions in these very different parasites. However, given the currently poor understanding of exactly what does K13 do in *P. falciparum* and also how K13 mutations confer resistance/delayed clearance, the conclusions of functional equivalence remain overly speculative.

Since our previous submission, additional evidence regarding the function of K13 has come to light. We have updated the discussion of this manuscript to reflect this new evidence, and hopefully provided a measured interpretation of our results regarding K13.

3. WGS of the DegP2 might answer what other genetic changes occurred in generating this mutant and provide additional understanding of genetic factors that can be linked to altered DHA sensitivity phenotypes.

This is a good point, but we chose to instead address these concerns by constructing DegP2 cKD, as discussed above, offering a more controlled way of examining the function of DegP2.

4. What is the % coverage of the genome by the CRISPR mutagenesis method used.

We targeted 97% of the genes in the *T. gondii* genome in these screens. Genes that were not targeted were limited to those that were necessary for selectable markers to function and genes that are too poorly annotated to target reliably, usually found in repetitive regions of the genome. In contrast to chemical mutagenesis, CRISPR mutagenesis introduces frameshift mutations, or leads to the integration of large DNA fragments in wild-type *T. gondii* (Sidik et al. 2014. *PLoS One*). We targeted the majority of genes with 10 gRNAs each, meaning that we induced mutations at 10 separate locations. A small proportion of genes were too short to target at 10 locations. Further details on the library construction and validation are available in the original publication (Sidik, Huet et al. 2016. *Cell*). We have added the following language to the manuscript in an attempt to briefly explain the technology.

“Transfecting a library containing 10 guide RNAs (gRNAs) per gene into a large population of parasites that constitutively expressed the Cas9 nuclease we created a diverse population of mutants. From previous work, we know that parasites acquire on average a single gRNA that directs Cas9 to create a double-stranded break in the coding sequence of the specified gene^{33,34,43}. Insertions and deletions introduced during DNA repair lead to loss-of-function mutations in the targeted genes, and the prevalence

of different mutants in the population can be inferred from the relative abundance of gRNAs against each gene.”

5. The manuscript uses a lot of technical jargon that lacks clear meaning for those not in this field of study (for example, “guide RNAs... were enriched”).

We have added the language stated above in an attempt to further clarify the technical aspects of the work.

Reviewers' Comments:

Reviewer #2:

Remarks to the Author:

The major contribution of this manuscript is its identification of novel genetic loci that modulate artemisinin (ART) sensitivity in *T. gondii* and demonstration that the *P. falciparum* ortholog of one of these genes (DegP2) influences artemisinin sensitivity in malaria parasites. Given the central importance of ART-based therapies in malaria treatment and their exploration for treating cancer and other infectious diseases, increased understanding of cellular features that influence ART sensitivity in diverse organisms is important, timely, and likely to impact on-going drug development/discovery.

The authors have made extensive efforts to address the prior critiques. Although the manuscript has been substantially improved by these revisions, I have several remaining suggestions and concerns that can likely be addressed by textual changes.

1. Title and abstract: Since heme biosynthesis is not a dominant determinant of ART sensitivity in *P. falciparum*, I suggest that the authors replace "heme biosynthesis" in the title and "the heme biosynthetic pathway" in the last line of the abstract with "heme metabolism" to make the title and abstract general to both organisms studied in this manuscript.
2. It still seems odd in the Introduction to transition directly from discussion of Hb endocytosis as a strong determinant of ART activation in *P. falciparum* to proposing a genetic screen in *T. gondii* to discover drug sensitivity mutations without mentioning that Hb import is not a feature of *T. gondii* biology. To address this issue fairly, I encourage the authors to explicitly state this difference while also perhaps noting that broad features of cellular and organelle biology are conserved and that studies with the more genetically tractable *T. gondii* can uncover novel ART sensitivity determinants beyond Hb import relevant for both parasites.
3. Prior work in cancer cells has suggested that mitochondrial heme synthesis is a major ART sensitivity determinant (e.g., <https://www.ncbi.nlm.nih.gov/pmc/articles/PMC2764339/>). Based on this precedent and since *T. gondii* lacks Hb uptake, a reasonable starting hypothesis (and what this reviewer had assumed but without any data) is that heme biosynthesis is critical for ART activity in *T. gondii*. It is reassuring that the authors observe heme synthesis mutations that provide ART resistance in *T. gondii*, even if unable to directly test with individual knock-down/knock-out parasites. The authors may wish to discuss these parallels.
4. In line 79, it seems odd to propose to "establish the susceptibility of *T. gondii* to DHA" without citing prior work indicating modest activity of ART and derivatives against *T. gondii* (e.g., <https://www.ncbi.nlm.nih.gov/pmc/articles/PMC6093624/> or earlier studies cited therein), for example in line 85.
5. Line 106: Is there a missing verb, e.g. "were"?
6. The authors have substantially revised the discussion to offer several hypotheses regarding possible mechanistic links between the gene mutations that alter ART sensitivity and variable porphyrin levels that accompany mutations. While these speculations are useful and appropriate for a Discussion section, the modest correlation between porphyrin levels (indirect estimate of heme) and varying ART sensitivity does not uniquely establish causation, especially given observations with ALA that defy this correlation. Thus, statements such as line 227 that the authors have "confirmed that modulating heme levels modulates DHA sensitivity" seem too dogmatic even if chemically reasonable as a hypothesis. I would encourage the authors to word their conclusions carefully. I thought the section heading in line

226 was a good example of a fair statement of one such connection.

7. In Figure 5e, the x-axis values of data points do not appear to correspond to the mean values depicted in Fig. 5c. For example, the mean porphyrin level in Δ DegP2 appear to be <50% of the parental in 5c, yet the data point for Δ DegP2 in 5e is given as >50%. The mean porphyrin levels for Δ DegP2/DegP2-HA in Fig. 5c and 5e also appear to differ. Can the authors clarify why these values are different in the 2 plots and what the correct values should be?

8. Fig. 5e is interesting, and the strength of this correlation (whether linear or non-linear) is critical for supporting the authors' hypothesis that variable porphyrin (heme) levels are the key mechanistic link between gene mutations and altered DHA sensitivity. To more broadly test this correlation, can the authors add the data points for drug treatments (e.g., SA, NaFAC, and 2-DG)? The ALA data will defy this correlation (and contrasts with prior observations in cancer cells- see reference in #4) but as the authors correctly point out ALA is the only treatment expected to cause accumulation of PPIX such that total porphyrin levels may inaccurately reflect total heme.

9. Lines 362-363: "Taken together, these data also demonstrate that screens in *T. gondii* can identify resistance alleles that act through mechanisms conserved across the apicomplexan phylum." I agree that the screens have identified a conserved resistance allele, but it is not clear to me that the mechanisms of resistance are conserved. The authors suggest in the Discussion that reports of mitochondrial K13 localization upon DHA treatment in *P. falciparum* could link DegP2 to Hb uptake, but such connections remain speculative. I suggest that the authors remove "that act through mechanisms".

Reviewer #3:

Remarks to the Author:

The objective of the study is to identify factors associated the susceptibility of *Toxoplasma gondii* to artemisinin as a model to decipher the mechanism of action of this main antimalarial drug in *Plasmodium falciparum*. The study uses an innovative genome-wide forward genetic screening approach to identify mutations in *T. gondii* that alter its artemisinin susceptibility to identify potential mechanisms that may be relevant for susceptibility in malaria parasites. The innovative approach and significance of the topic elevate the significance of the outcomes of the study.

Both *T. gondii* and *P. falciparum* are major disease-causing Apicomplexa, have similar intracellular life cycles, and are sensitive to artemisinin treatment (albeit with a 100-fold difference in efficacy). Nonetheless, there are significant differences in the species-specific biology for how useful this model might be. This was a major concern in the original manuscript and the authors have revised the manuscript in response to all the reviewers' critiques. Some additional qualifications about the limits of the model and the respective differences in basic biology were added to put the outcomes of the study in a better perspective. Clearly, there are many additional unknown unknowns – this is true for model organism studies – but in regards to this study, the potential for false positives is acceptable considering the overall conclusion is consistent with our current understanding of artemisinin MOA/mechanisms of susceptibility. Since the original submission, new studies on K13's biological function have greatly enhanced our understanding about its role and how perturbation of this function can alter *P. falciparum* metabolic processes and artemisinin susceptibility. Despite the significant new knowledge gained, it is quite clear this remains only part of the story and does not conflict with the conclusions of the conclusions of this manuscript. The current study leverages the advantages of the model system in an approach still very challenging to do in *P. falciparum* to reveal metabolic processes that might be related to artemisinin MOA. Included are additional experiments to provide validation in

P. falciparum and support the conclusions. However, it is obvious additional follow-on studies are needed to truly appreciate the significance of these findings.

The authors have responded adequately to my major concerns for the original manuscript. Creation of an inducible DegP2 mutant did confirm the growth defects of the original mutant strain were not related to the loss of this product. The added caveats for inclusion of all 73 genes putatively involved in regulating DHA susceptibility are partially adequate. Not included are additional relevant functional data available from whole genome screens of *P. berghei* and *P. falciparum* as well as different 'omics analyses. The justification for the broad inclusion of all these pathways remains weak and these previous studies could be cited to support the significance the pathway analysis. Finally, the authors' message is clearer, although an additional qualification should be noted that a significant result of a new genetic cross was the in vitro RSA was not consistent of in vivo outcomes. The additional edits in response to my minor concerns have improved clarity of the manuscript and understanding of the significance of their conclusions.

REVIEWER COMMENTS

*Author responses highlighted in blue

Reviewer #2 (Remarks to the Author):

The major contribution of this manuscript is its identification of novel genetic loci that modulate artemisinin (ART) sensitivity in *T. gondii* and demonstration that the *P. falciparum* ortholog of one of these genes (DegP2) influences artemisinin sensitivity in malaria parasites. Given the central importance of ART-based therapies in malaria treatment and their exploration for treating cancer and other infectious diseases, increased understanding of cellular features that influence ART sensitivity in diverse organisms is important, timely, and likely to impact on-going drug development/discovery.

The authors have made extensive efforts to address the prior critiques. Although the manuscript has been substantially improved by these revisions, I have several remaining suggestions and concerns that can likely be addressed by textual changes.

1. Title and abstract: Since heme biosynthesis is not a dominant determinant of ART sensitivity in *P. falciparum*, I suggest that the authors replace “heme biosynthesis” in the title and “the heme biosynthetic pathway” in the last line of the abstract with “heme metabolism” to make the title and abstract general to both organisms studied in this manuscript.

The title and abstract have been modified at the reviewer’s request.

2. It still seems odd in the Introduction to transition directly from discussion of Hb endocytosis as a strong determinant of ART activation in *P. falciparum* to proposing a genetic screen in *T. gondii* to discover drug sensitivity mutations without mentioning that Hb import is not a feature of *T. gondii* biology. To address this issue fairly, I encourage the authors to explicitly state this difference while also perhaps noting that broad features of cellular and organelle biology are conserved and that studies with the more genetically tractable *T. gondii* can uncover novel ART sensitivity determinants beyond Hb import relevant for both parasites.

Following the reviewer’s advice we have more explicitly noted this in the introduction, stating:

“Despite species-specific differences that could impact artemisinin susceptibility—such as the lack of substantial hemoglobin uptake by *T. gondii*—Here we demonstrate that a point mutation in K13, homologous to the canonical *P. falciparum* K13C580Y, reduced the susceptibility of *T. gondii* to DHA.”

While we do mention that hemoglobin endocytosis is an important determinant of artemisinin susceptibility, we also discuss other potential contributing pathways that have not been entirely explained. Since the point of the screens is to examine the entire genome for such chemical-genetic interactions, we did not have any preconceived notions of what we might find.

3. Prior work in cancer cells has suggested that mitochondrial heme synthesis is a major ART sensitivity determinant (e.g., <https://www.ncbi.nlm.nih.gov/pmc/articles/PMC2764339/>). Based on this precedent and since *T. gondii* lacks Hb uptake, a reasonable starting hypothesis (and what this reviewer had assumed but without any data) is that heme biosynthesis is critical for ART activity in *T. gondii*. It is reassuring that the authors observe heme synthesis mutations that provide ART resistance in *T. gondii*, even if unable to directly test with individual knock-down/knock-out parasites. The authors may wish to discuss these parallels.

We are glad our experimental observations conform to the reviewer's assumptions. Indeed we have already cited the study by Zhang and Gerhard (54) in the discussion. We have more explicitly stated that connection:

“Modulation of heme biosynthesis in cancer cells has similarly been found to alter their susceptibility to artemisinin(Zhang and Gerhard 2009).”

4. In line 79, it seems odd to propose to “establish the susceptibility of *T. gondii* to DHA” without citing prior work indicating modest activity of ART and derivatives against *T. gondii* (e.g., <https://www.ncbi.nlm.nih.gov/pmc/articles/PMC6093624/> or earlier studies cited therein), for example in line 85.

We agree with the reviewer that it is important for us to qualify the statement. We now introduce this section of the results with the following sentence:

“Previous studies have demonstrated the susceptibility of *T. gondii* to artemisinin and its derivatives (Radke et al. 2018; Nagamune et al. 2007; Dunay et al. 2009).”

We also qualify that the intention was to establish susceptibility “in our assays” as follows:

“To establish the susceptibility of *T. gondii* to DHA in our assays...”

5. Line 106: Is there a missing verb, e.g. “were”?

Thank you for pointing it out. It has been corrected.

6. The authors have substantially revised the discussion to offer several hypotheses regarding possible mechanistic links between the gene mutations that alter ART sensitivity and variable porphyrin levels that accompany mutations. While these speculations are useful and appropriate for a Discussion section, the modest correlation between porphyrin levels (indirect estimate of heme) and varying ART sensitivity does not uniquely establish causation, especially given observations with ALA that defy this correlation. Thus, statements such as line 227 that the authors have “confirmed that modulating heme levels modulates DHA sensitivity” seem too dogmatic even if chemically reasonable as a hypothesis. I would encourage the authors to word their conclusions carefully. I thought the section heading in line 226 was a good example of a fair statement of one such connection.

In other settings, the genetic results of the screen, validated through the use of two inhibitors, might have been considered sufficient to establish causality. In the previous draft of the manuscript, we had shown that the ALA result is in fact not contradictory; by mass spectrometry, ALA could be shown to increase the levels of PPIX but not heme.

Unfortunately, due to the expense and challenge of repeating the metabolomics experiments, those data were removed from the final draft. Nevertheless, to satisfy the reviewer's request, we have revised the relevant passage to more modestly state the conclusion:

“Having verified the relationship between heme biosynthesis and DHA susceptibility...”

7. In Figure 5e, the x-axis values of data points do not appear to correspond to the mean values depicted in Fig. 5c. For example, the mean porphyrin level in Δ DegP2 appear to be <50% of the parental in 5c, yet the data point for Δ DegP2 in 5e is given as >50%. The mean

porphyrin levels for Δ DegP2/DegP2-HA in Fig. 5c and 5e also appear to differ. Can the authors clarify why these values are different in the 2 plots and what the correct values should be?

During the preparation of the previous revision, we had failed to replace Figure 5e with one prepared using the absolute porphyrin measurements. We have revised the figure using the new values which now precisely correspond to the preceding figures.

8. Fig. 5e is interesting, and the strength of this correlation (whether linear or non-linear) is critical for supporting the authors' hypothesis that variable porphyrin (heme) levels are the key mechanistic link between gene mutations and altered DHA sensitivity. To more broadly test this correlation, can the authors add the data points for drug treatments (e.g., SA, NaFAC, and 2-DG)? The ALA data will defy this correlation (and contrasts with prior observations in cancer cells- see reference in #4) but as the authors correctly point out ALA is the only treatment expected to cause accumulation of PPIX such that total porphyrin levels may inaccurately reflect total heme.

Following the reviewer's suggestion we incorporated the results from SA and NaFAC treatment and they conform nicely to the negative correlation highlighted in Fig. 5e. Based on the results described above for ALA, and the reviewer's comment, we excluded ALA treatment from this correlation.

9. Lines 362-363: "Taken together, these data also demonstrate that screens in *T. gondii* can identify resistance alleles that act through mechanisms conserved across the apicomplexan phylum." I agree that the screens have identified a conserved resistance allele, but it is not clear to me that the mechanisms of resistance are conserved. The authors suggest in the Discussion that reports of mitochondrial K13 localization upon DHA treatment in *P. falciparum* could link DegP2 to Hb uptake, but such connections remain speculative. I suggest that the authors remove "that act through mechanisms".

The mechanism specified is the alteration in heme availability, which we demonstrate for knockout and depletion of DegP2 in *T. gondii* and knockout of DegP in *P. falciparum*. Based on the extensive literature linking heme to the activation of DHA, we believe that this constitutes a common mechanism. Furthermore, as far as we can tell the K13 and DegP/DegP2 alleles are orthologous, such that conserved mechanisms of resistance are the most parsimonious explanation of the data. We therefore argue to retain the present wording.

Reviewer #3 (Remarks to the Author):

The objective of the study is to identify factors associated with the susceptibility of *Toxoplasma gondii* to artemisinin as a model to decipher the mechanism of action of this main antimalarial drug in *Plasmodium falciparum*. The study uses an innovative genome-wide forward genetic screening approach to identify mutations in *T. gondii* that alter its artemisinin susceptibility to identify potential mechanisms that may be relevant for susceptibility in malaria parasites. The innovative approach and significance of the topic elevate the significance of the outcomes of the study.

Both *T. gondii* and *P. falciparum* are major disease-causing Apicomplexa, have similar intracellular life cycles, and are sensitive to artemisinin treatment (albeit with a 100-fold difference in efficacy). Nonetheless, there are significant differences in the species-specific biology for how useful this model might be. This was a major concern in the original manuscript and the authors have revised the manuscript in response to all the reviewers' critiques. Some additional qualifications about the limits of the model and the respective differences in basic biology were added to put the outcomes of the study in a better perspective. Clearly, there are many additional unknown unknowns – this is true for model organism studies – but in regards to this study, the potential for false positives is acceptable considering the overall

conclusion is consistent with our current understanding of artemisinin MOA/mechanisms of susceptibility. Since the original submission, new studies on K13's biological function have greatly enhanced our understanding about its role and how perturbation of this function can alter *P. falciparum* metabolic processes and artemisinin susceptibility. Despite the significant new knowledge gained, it is quite clear this remains only part of the story and does not conflict with the conclusions of this manuscript. The current study leverages the advantages of the model system in an approach still very challenging to do in *P. falciparum* to reveal metabolic processes that might be related to artemisinin MOA. Included are additional experiments to provide validation in *P. falciparum* and support the conclusions. However, it is obvious additional follow-on studies are needed to truly appreciate the significance of these findings.

The authors have responded adequately to my major concerns for the original manuscript. Creation of an inducible DegP2 mutant did confirm the growth defects of the original mutant strain were not related to the loss of this product. The added caveats for inclusion of all 73 genes putatively involved in regulating DHA susceptibility are partially adequate. Not included are additional relevant functional data available from whole genome screens of *P. berghei* and *P. falciparum* as well as different 'omics analyses. The justification for the broad inclusion of all these pathways remains weak and these previous studies could be cited to support the significance the pathway analysis. Finally, the authors' message is clearer, although an additional qualification should be noted that a significant result of a new genetic cross was the *in vitro* RSA was not consistent of *in vivo* outcomes. The additional edits in response to my minor concerns have improved clarity of the manuscript and understanding of the significance of their conclusions.

We appreciate the reviewer's response to our revised manuscript. We agree that there are many unanswered questions beyond the scope of our study. We cannot identify which relevant functional data could have enhanced the manuscript, but hope that such metanalyses will be accessible to others upon the publication of our work.